# Phloem unloading in Arabidopsis roots is convective and regulated by the phloem-pole pericycle

Timothy J Ross-Elliott[1], Kaare H Jensen[2], Katrine S Haaning[2], Brittney M Wager[1], Jan Knoblauch[1], Alexander H Howell[1], Daniel L Mullendore[1], Alexander G Monteith[3], Danae Paultre[4], Dawei Yan[5], Sofia Otero[5], Matthieu Bourdon[5], Ross Sager[6], Jung-Youn Lee[6], Ykä Helariutta[5], Michael Knoblauch[1]*, Karl J Oparka[4]*

[1]School of Biological Sciences, Washington State University, Pullman, United States; [2]Department of Physics, Technical University of Denmark, Lyngby, Denmark; [3]Department of Biological and Medical Sciences, Oxford Brookes University, Headington, Oxford, United Kingdom; [4]Institute of Molecular Plant Science, University of Edinburgh, Edinburgh, United Kingdom; [5]Sainsbury Laboratory, University of Cambridge, Cambridge, United Kingdom; [6]Department of Plant and Soil Sciences, Delaware Biotechnology Institute, University of Delaware, Newark, United States

*For correspondence:
knoblauch@wsu.edu (MK); karl.
oparka@ed.ac.uk (KJO)

**Competing interests:** The authors declare that no competing interests exist.

**Abstract** In plants, a complex mixture of solutes and macromolecules is transported by the phloem. Here, we examined how solutes and macromolecules are separated when they exit the phloem during the unloading process. We used a combination of approaches (non-invasive imaging, 3D-electron microscopy, and mathematical modelling) to show that phloem unloading of solutes in Arabidopsis roots occurs through plasmodesmata by a combination of mass flow and diffusion (convective phloem unloading). During unloading, solutes and proteins are diverted into the phloem-pole pericycle, a tissue connected to the protophloem by a unique class of 'funnel plasmodesmata'. While solutes are unloaded without restriction, large proteins are released through funnel plasmodesmata in discrete pulses, a phenomenon we refer to as 'batch unloading'. Unlike solutes, these proteins remain restricted to the phloem-pole pericycle. Our data demonstrate a major role for the phloem-pole pericycle in regulating phloem unloading in roots.

## Introduction

In plants, the products of photosynthesis in green tissues are delivered by the phloem to distant organs where they are utilized in growth or storage (*Turgeon and Wolf, 2009*; *De Schepper et al., 2013*). Modern agriculture aims to maximize the amounts of carbon-based products allocated to storage organs such as grains and tubers, structures that act as carbon 'sinks' for the assimilates delivered by the phloem. The process by which solutes exit the phloem is termed phloem unloading, and is a central target for regulating the flux of carbon into sink tissues (*Ham and Lucas, 2014*). In addition to assimilates, phloem sap also contains numerous proteins and RNAs (*Kehr, 2006*; *Atkins et al., 2011*; *Batailler et al., 2012*; *Turnbull and Lopez-Cobollo, 2013*). Unloading must therefore combine the seemingly antagonist functions of high selectivity with large permeability in order to allocate assimilates to growth zones while controlling the movement of macromolecules.

**eLife digest** A mechanism called photosynthesis allows plants to use energy from sunlight to make sugars from carbon dioxide gas and water. These sugars can then be used as fuel, or as building blocks for wood and other plant structures. Every part of the plant requires sugars, but most photosynthesis happens in the leaves and stems, so the sugars need to be able to move around the plant to wherever they are needed.

Phloem tubes form a network that transports sugar, proteins and other molecules around the plant within a fluid known as sap. Because this network is so extensive, it is very difficult to study, which has left researchers with major questions about how it works. For example, it is not clear how the sugar and other molecules leave the phloem when they reach their destination.

Ross-Elliot et al. used a combination of microscopy and mathematical modeling to investigate how sugars and other molecules leave the phloem in the roots of a plant called *Arabidopsis thaliana*. The experiments show that these molecules move directly into cells within a neighboring tissue called the phloem-pole pericycle via pores known as funnel plasmodesmata.

Ross-Elliot et al. incorporated the experimental data into a mathematical model of phloem unloading. This model suggests that sugars and other small molecules move freely through the funnel plasmodesmata, but large proteins pass through these pores in pulses. Future challenges include finding out exactly how plants control phloem unloading and to investigate whether it is possible to modify the delivery of specific molecules to particular parts of the plant.

The mechanisms that enable the precise and coordinated unloading of phloem-mobile compounds remain unknown.

Solute flow occurs through sieve elements (SEs), elongated cells connected to each other by perforated end walls called sieve plates. During differentiation SEs lose most of their cellular components, including their nucleus (*Oparka and Turgeon, 1999*; *Furuta et al., 2014*). Ontogenetically related, metabolically active companion cells (CCs) support the adjacent enucleate SEs throughout their lifespan (*Lucas et al., 1996*; *Pritchard, 1996*; *van Bel and Knoblauch, 2000*; *Lalonde et al., 2001*; *Otero and Helariutta, 2017*). The pressure flow model of phloem transport, originally proposed by *Münch (1930)*, envisages that an osmotically generated pressure differential drives the bulk flow through the SEs that connect photosynthetic tissues (sources) with those in which carbon consumption occurs (sinks). The loading of solutes in source tissues results in a high osmotic potential within SEs. The reduction of turgor pressure in sink organs due to carbon consumption leads to a pressure gradient that provides the energy to overcome viscous resistance within the SEs, resulting in a passive phloem flow from source to sink (*Münch, 1930*). Phloem sap collected from excised aphid stylets contains a complex mixture of macromolecules and low-molecular weight solutes (*Atkins et al., 2011*). In a recent study, *Paultre et al. (2016)* showed that many proteins, including those with targeting sequences, can move across a graft union and be unloaded near the root tip. Significantly, these proteins entered a post-phloem domain beyond which their movement was restricted. This observation begs the question as to how the phloem is able to discriminate between macromolecules and solutes during unloading. Besides the central function in resource allocation, it is now well established that the phloem also serves as network for transmission of chemical (*Kramer and Boyer, 1995a*; *Mullendore et al., 2015*) and electrical (*Hedrich et al., 2016*) signals.

Phloem unloading in actively growing tissues such as the shoot or root apex occurs through the protophloem, a transient tissue that connects the conducting phloem with the receiver cells in sink tissues (*Oparka et al., 1994*). Initial investigations of phloem unloading in the root tip of Arabidopsis (*Oparka et al., 1994*, *1995*; *Wright and Oparka, 1996*) provided evidence that unloading occurs through plasmodesmata (PD), the specialized pores that connect plant cells. Due to cell division and growth in the apical region of the root the demand for assimilates is high. Protophloem sieve elements (PSEs) become mature in such regions to accommodate this demand while the neighboring cells are still differentiating (*Furuta et al., 2014*). Because PSEs lose their nucleus, they cannot divide to keep pace with the growth of the neighboring cells. Therefore, they become increasingly elongated until they become inactive in transport and eventually obliterated (*Erwin and Evert, 1967*;

*Eleftheriou and Tsekos, 1982*; *Furuta et al., 2014*). In the elongation zone of the root, solutes are transferred laterally from the metaphloem SEs (MSEs) to the PSEs, allowing phloem continuity between source and sink tissues (*Stadler et al., 2005*; *Winter et al., 1992*).

A currently favored hypothesis of phloem unloading is the 'high-pressure manifold model' proposed by *Fisher (2000)*, recently evaluated by *Patrick (2013)*. A central element of this model is that a low pressure gradient occurs along the flow path, with a steep drop in pressure between the PSEs and surrounding cells in the phloem unloading zone. In this scenario, allocation of carbon is controlled by the lateral hydraulic conductance in the unloading zone. Recent phloem turgor measurements in morning glory, however, did not support this model, as the bulk of the pressure is consumed by friction within the SEs, and only small pressure gradients are available for unloading (*Knoblauch et al., 2016*). The *Fisher (2000)* model also does not explain how both small solutes and macromolecules can leave the phloem simultaneously. *Paultre et al., 2016* suggested recently that the removal of mobile proteins into a post-phloem domain may be necessary to prevent the terminal PSEs from becoming occluded, an event that would lead to dissipation of the turgor gradient between source and sink. Unfortunately, many of the factors that determine phloem unloading are not well studied. This is because the phloem in most sinks is difficult to access as it is embedded in an opaque layer of tissue. Phloem transport ceases immediately when the source is detached from the sink. Therefore, the function of the phloem can only be studied in situ, requiring new approaches for dissecting the factors that regulate phloem unloading.

The many unknowns surrounding phloem unloading in plants prompted us to conduct a detailed structure/function study of the terminal PSEs in the phloem unloading zone which, in Arabidopsis, is amenable to non-invasive imaging (*Oparka et al., 1994*; *Knoblauch et al., 2015*). We combined a detailed ultrastructural analysis of the cellular interfaces in the phloem unloading zone with the kinetics of phloem unloading of fluorescent solutes and macromolecules obtained by real-time imaging of growing roots. This analysis allowed us to derive new quantitative data on the factors that regulate phloem unloading in Arabidopsis. We report on the presence of a unique class of 'funnel plasmodesmata' that are involved specifically in the unloading of molecules into the phloem-pole pericycle. We show, by mathematical modelling, that phloem unloading of small solutes from PSEs is convective, i.e. it occurs continuously by a combination of mass flow and diffusion. In contrast, we find that macromolecules are unloaded in discrete pulses, a phenomenon we refer to as 'batch unloading'. These macromolecules are diverted specifically into the two phloem-pole pericycle cells that abut each PSE where they are filtered out from the unloaded solutes.

## Results

### Calculating the dimensions of the phloem unloading zone using phloem-mobile probes

In Arabidopsis roots, phloem unloading occurs exclusively from the protophloem, a short-lived tissue that is functional in the zone of root elongation (*Oparka et al., 1994*). Published images of phloem unloading in Arabidopsis give the impression that the unloading zone is a relatively broad region (*Oparka et al., 1994*; *Wright and Oparka, 1996*; *Knoblauch et al., 2015*). However, such images represent a 'snapshot' of unloading, taken at a defined time point following the application of fluorescent solutes to the leaf. Carboxyfluorescein diacetate (CFDA) is the most widely utilized phloem-mobile probe (*Oparka et al., 1994*; *Wright and Oparka, 1996*; *Knoblauch and van Bel, 1998*). This probe is non-fluorescent when applied to source leaves but is subsequently cleaved by endogenous esterases to produce fluorescent, membrane-impermeant, carboxyfluorescein (CF; *Knoblauch et al., 2015*). The dye travels with the phloem translocation stream to sink tissues, where it can be visualized (*Figure 1A*). When studied in real time, the dye characteristically shows preferential movement outwards into the cortex relative to the stele (*Figure 1A*; see also *Oparka et al., 1994*). Unloading of the dye in the root tip indicates a symplastic (plasmodesmata-mediated) pathway (*Oparka et al., 1994*; *Wright and Oparka, 1996*; *Knoblauch et al., 2015*). The cells in the unloading zone sequester the dye rapidly into their vacuoles where it becomes trapped. This results in an increasing fluorescence within cells over time (*Wright and Oparka, 1996*). Due to root growth, the cells initially involved in unloading move out of the phloem-unloading zone basipetally, but maintain their fluorescence due to the presence of dye in their vacuoles. Therefore, the apparent

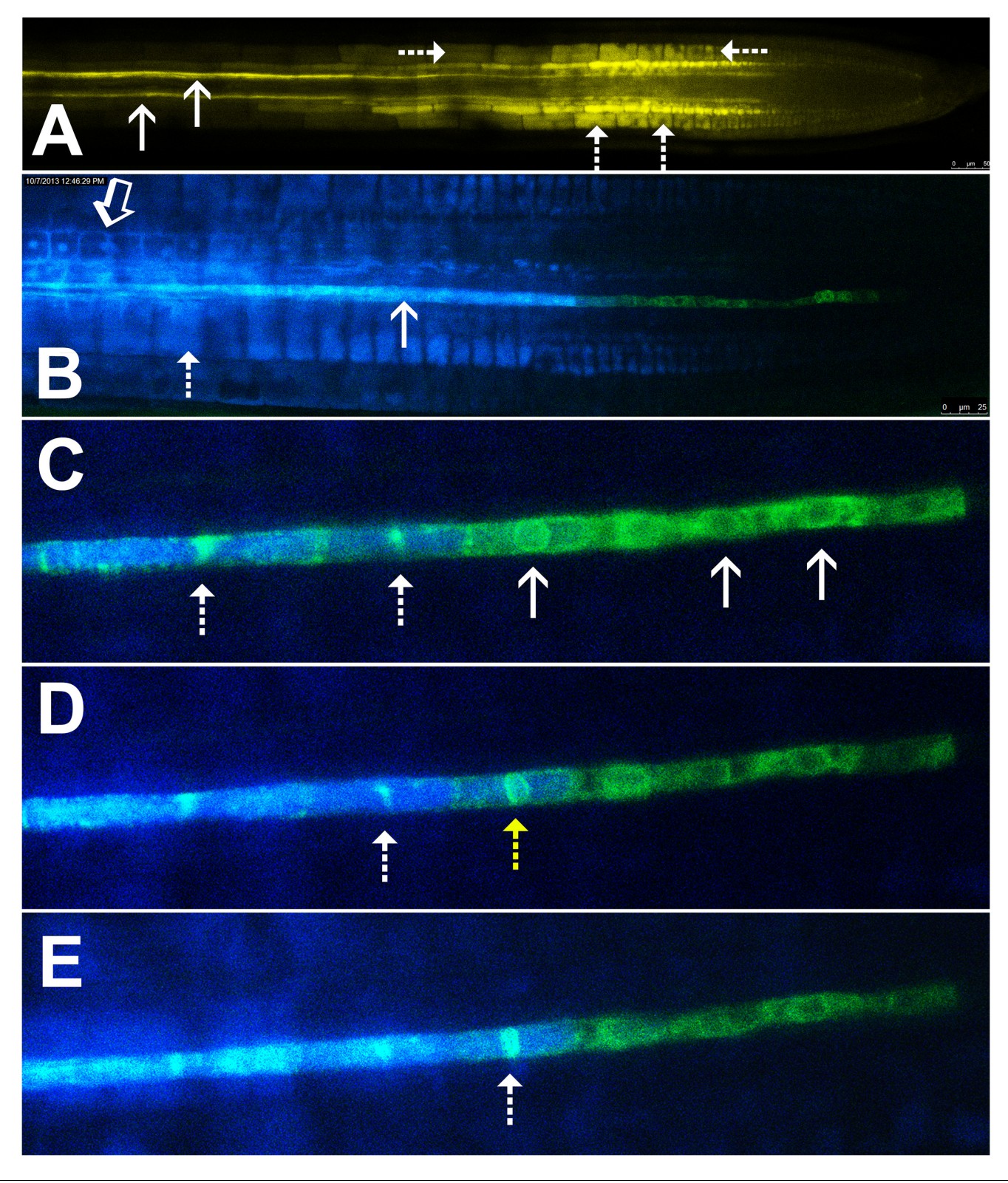

**Figure 1.** Symplastic unloading of phloem mobile probes. (**A**) 2D optical section of unloading of CFDA in the root tip. The two protophloem files leading into the root tip are shown (solid arrows) and sequestration of CFDA into the vacuoles is apparent (dashed arrows). (**B**) Unloading of esculin (blue) in the root tip of a transgenic Arabidopsis line expressing GFP (green) targeted to the ER lumen of the PSE (pMtSEO2::GFP5-ER). Esculin escapes the protophloem file (solid arrow) into the cytoplasm of neighboring cells (open arrow). In contrast to CFDA, esculin is only sequestered in the vacuoles

*Figure 1 continued on next page*

*Figure 1 continued*

at high concentrations (dashed arrow). (**C–E**) Three frames extracted from *Video 1*. (**C**) GFP targeted to the ER lumen of PSEs demarcates the nuclear membrane of young sieve elements that have not yet been integrated into the unloading zone (solid arrows). Dashed arrows indicate two degrading nuclei in cells that are already filled with esculin (blue) (also for **D** and **E**). (**D**) Degradation of the nucleus (yellow arrow) coincides with the opening of the sieve-plate pores, allowing esculin (blue) to enter the cell. This defines the new PSE zero. (**E**) As nuclear degradation continues, the sieve element becomes an integral member of the phloem unloading zone.

unloading zone broadens over time as root growth progresses, obscuring the current site of unloading. We therefore sought to define precisely the dimensions of the true phloem-unloading zone.

Esculin is a naturally fluorescent, glucosylated coumarin derivative recently introduced as phloem-mobile probe (*Knoblauch et al., 2015*). Unlike CFDA, it is loaded into the phloem by the sucrose transporter, SUC2, in the CCs of source tissues (*Gora et al., 2012*; *Knoblauch et al., 2015*). Sequestration of this probe is minimal and occurs only when high concentrations of the probe are applied (*Figure 1B*). This feature allows for extended acquisition of time-lapse movies. Unlike CF, esculin can also be detected clearly in lines expressing GFP (*Knoblauch et al., 2015*).

To define the developmental stage at which differentiating PSEs become integrated into the unloading zone, we used transgenic Arabidopsis lines expressing GFP targeted to the sieve-element ER (HDEL-GFP) under control of the MtSEO2 promoter, a SE-specific promoter (*Froelich et al., 2011*; *Knoblauch and Peters, 2010*); *Figure 1B–E*). This transgenic line clearly demarcates PSEs in the early stages of differentiation and is an excellent marker of the nuclear membrane (*Figure 1C–E*). We loaded source leaves of this line with esculin and acquired time-lapse movies of unloading in the terminal PSEs. Esculin did not enter differentiating PSEs that were symplastically isolated from the rest of the protophloem file. These cells still had a fully intact nuclear membrane (*Figure 1C*, *Video 1*). Once the nuclear membrane started to degenerate the sieve-plate pores opened rapidly, allowing esculin to enter the cell, which then became an integral member of the protophloem-unloading zone (*Figure 1D and E*). For convenience of reference, we refer to this protophloem cell as 'PSE zero' (yellow dashed arrow in *Figure 1D*). Remnants of the nuclear membrane remained for some time in PSEs that had been newly incorporated into the phloem-unloading zone (*Figure 1C–E*). These observations can be seen in *Video 1*. Our data suggest that nucleate, differentiating PSEs are isolated from the translocation stream. However, the degeneration of the nucleus and the opening of the sieve-plate pores are closely related events that lead to incorporation of PSE zero into the phloem-unloading zone. At this point the cell becomes competent to unload solutes (*Figure 1C–E*; *Video 1*).

In order to acquire functional data on the dimensions of the unloading zone, we conducted flow velocity measurements in individual PSE files using fluorescence recovery after photobleaching (FRAP; *Froelich et al., 2011*). Phloem flow velocities in the terminal region of the root (basipetal to the unloading zone) are in the range of 25 µm/s (*Froelich et al., 2011*). In this study, we photobleached CF after it had arrived in the phloem unloading zone of the root. In a tube of constant diameter and impermeable walls, the flow velocity is constant. When the walls become leaky, the flow velocity decreases because of loss of fluid. In the case of the protophloem, the tube becomes leaky when unloading occurs. To define the size of the unloading zone, we measured flow velocity along the phloem files and found that deceleration started at about 300–400 µm behind PSE zero (this dimension varied slightly, even within the two sieve tube files in the same root tip; *Figure 2*). Our data revealed that the phloem of the root is subdivided into distinct structural and functional domains. Translocation into the main root occurs through the metaphloem. In the elongation zone of the root, the metaphloem overlaps with the mature protophloem file, at which point solutes are

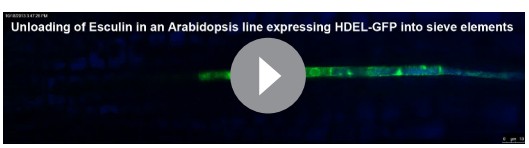

**Video 1.** Visualization of the development of PSE zero. An Arabidopsis line with GFP tagged ER (green) in the protophloem sieve elements is loaded with Esculin (blue). When the nucleus in the sieve element degrades, sieve plate pores open and the blue Esculin enters the cell. This integrates the cell into the unloading zone and defines a new PSE zero.

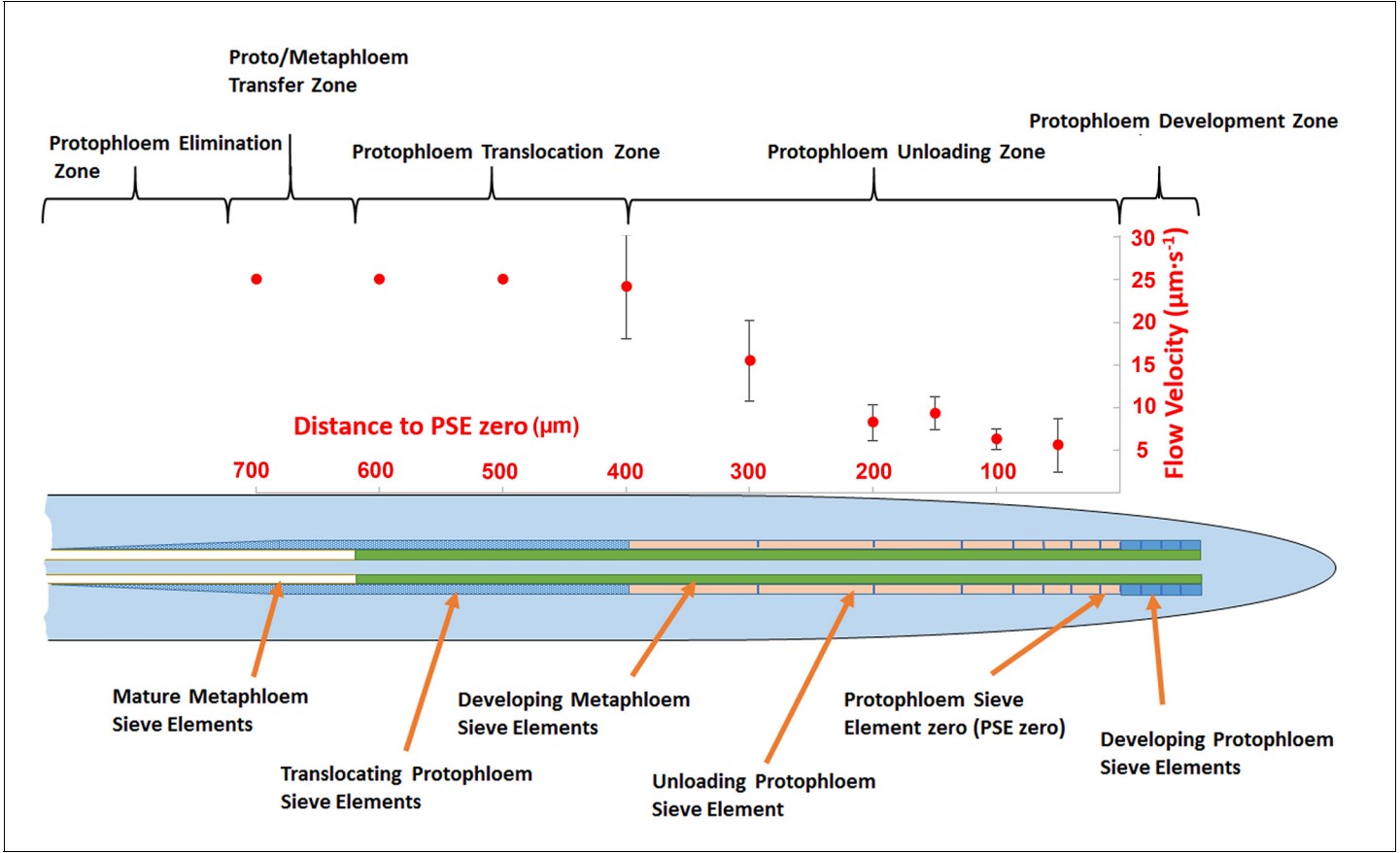

**Figure 2.** Schematic diagram illustrating the organization of phloem cells in specific zones in the root tip of Arabidopsis. The graph represents experimentally derived velocities at defined points relative to the terminal sieve element (PSE zero) in the protophloem unloading zone. Error bars show standard deviation of the mean (n = 8).

transferred laterally from the metaphloem to the protophloem ('transfer zone'; *Figure 2*). Movement then occurs through the protophloem towards the root tip ('protophloem translocation zone'; *Figure 2*) and subsequently into the terminal functional PSEs ('protophloem unloading zone'), at which point solutes are distributed laterally into the root tip. Apical to this zone lies the 'protophloem differentiation zone' that, as described above, plays no role in phloem unloading (*Figure 2*).

## Callose deposition demarcates the protophloem translocation zone

Arabidopsis roots grow at speeds of about 100–150 µm per hour (*Beemster and Baskin, 1998*). Consequently, after the sieve-plate pores open, a single PSE in the phloem unloading zone is active in unloading for only about 2–4 hr. Initially, cellular remnants such as the nuclear membrane, ribosomes and tonoplast are degraded and removed from the PSE. Subsequently, the PSEs become physically stretched in the elongation zone and finally move into an area that is active in translocation but inactive in unloading ('protophloem translocation zone'; *Figure 2*). The PSEs that progress basipetally from the phloem unloading zone into the protophloem translocation zone must undergo a rapid cellular transformation, particularly in the PD on their lateral walls. However, the structural alterations that control the cessation of unloading are not known.

Callose is deposited in the neck region of plasmodesmata and restricts the size exclusion limit of the pore in response to various stimuli (*Luna et al., 2011*; *Nakashima et al., 2003*; *Radford et al., 1998*). To investigate if callose was involved in occluding PD in the translocation zone, we stained roots of intact seedlings with Sirofluor (*Evans et al., 1984*; *Vatén et al., 2011*). This fluorochrome has a strong affinity for callose and demarcates sieve plates and PD in developing cell walls (*Stone et al., 1984*; *Vatén et al., 2011*); *Figure 3*). The two protophloem files could be visualized

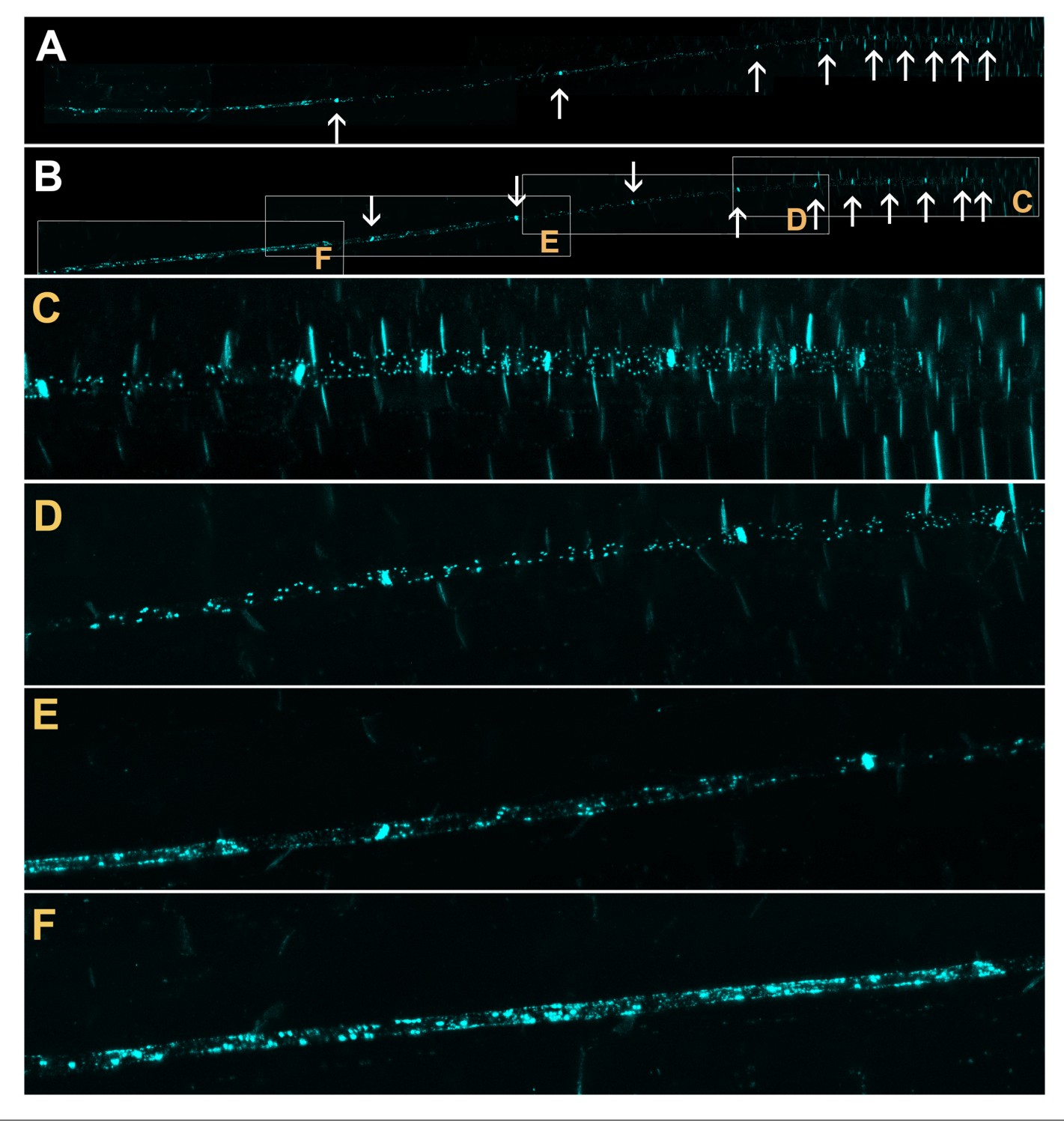

**Figure 3.** Confocal micrographs of the unloading zone in the Arabidopsis root tip stained with Sirofluor. Low magnification images showing the relatively strong fluorescence at sieve plates (arrows). (C–F) Higher magnification images at the locations indicated by boxes in (B). Individual plasmodesmata are resolved in the unloading zone (C, D). In the translocation zone, large deposits of callose are abundant (E, F).

easily with this stain (*Figure 3A and B*) and individual lateral PD were identified in the phloem unloading zone (*Figure 3C and D*). At the junction of the phloem unloading zone with the phloem translocation zone, callose deposits on the PSE wall increased in number (*Figure 3E*) until significant parts of the PSE walls were covered in callose (*Figure 3F*). This deposition of callose along the lateral walls of PSEs in the protophloem translocation zone correlated with the reduction in flow we observed in dye-loading experiments and may provide a structural basis for the functional isolation of this zone.

## Unloading of solutes and proteins involves a cellular division of labor

As well as sugars and amino acids, a large number of macromolecules including nucleic acids, and proteins occur in various amounts in the translocation stream (*Fukumorita and Chino, 1982*; *Kehr, 2006*; *Turnbull and Lopez-Cobollo, 2013*). Many of these may access the phloem sap via companion cells (CCs; *Paultre et al., 2016*). In addition, when new SEs develop in source regions of the plant, remnants of the nucleus, ribosomes, and tonoplast pass into the translocation stream. In both these scenarios, macromolecules must be removed from the terminal PSEs involved in unloading or they would impede the process of phloem unloading (*Paultre et al., 2016*). While small solutes and GFP (Stokes radius 2.82 nm; *Terry et al., 1995*) are unloaded from the PSEs into surrounding cells (*Imlau et al., 1999*), larger proteins unloaded from the PSE enter a unique post-phloem domain (*Stadler et al., 2005*; *Paultre et al., 2016*) and cannot cross the interface between the pericycle and endodermis (*Paultre et al., 2016*). We therefore sought to identify this domain as well as the routes taken by solutes and macromolecules during the unloading process.

In Arabidopsis, the root protophloem file is surrounded by five distinct cell files (*Figure 4A and B*). One immature metaphloem sieve element (MSE), located towards the center of the root, is in contact with the PSE. Two CCs, one to the left and one to the right, share a common cell wall area with the PSE and MSE, respectively. The complex is capped by two phloem pole pericycle (PPP) cell files (*Figure 4A and B*) that share cell walls with both the PSE and CCs. Small probes of the dimensions of sugars and amino acids (CFDA, esculin), and small proteins (GFP) move from cell to cell throughout the entire root tip without noticeable barriers (*Figure 1*; *Oparka et al., 1994*; *Stadler et al., 2005*; *Wright and Oparka, 1996*). However, translocated proteins of CC origin, are restricted to cells immediately adjacent to the mature protophloem file (*Stadler et al., 2005*; *Paultre et al., 2016*).

## The phloem-pole pericycle (PPP) is the repository for unloaded macromolecules

To monitor the fate of different phloem cargos, we compared the phloem-unloading pathway of small fluorescent probes (*Knoblauch et al., 2015*) with fluorescent fusion proteins (*Froelich et al., 2011*; *Stadler et al., 2005*), covering a molecular mass range of 340 Da to more than 100 kDa. Experiments were conducted in situ on living roots. As shown before, small probes were unloaded laterally towards the cortex and subsequently distributed amongst all root cells (*Figure 1*). Large probes of CC origin, however, were restricted to a specific domain (*Figures 4* and *5*). We wished to determine if proteins synthesized within SEs, rather than CCs, would show a similar pattern of unloading. Accordingly, we generated a transgenic Arabidopsis line expressing the Sieve Element Occlusion Related (SEOR) protein (*Froelich et al., 2011*; *Pélissier et al., 2008*) fused to YFP (SEOR-YFP). This protein is generated exclusively in young sieve elements and remains a structural component after maturation of the cell and its integration into the translocating sieve tube file. We crossed this line with a second reporter line in which HDEL-GFP was targeted to the ER (*Figure 1*). Both these fluorescent fusions were generated under the sieve-element specific promoters pSEOR; pMtSEO2. The promoters are active in young sieve elements only, but the gene products remain as structural components in mature sieve tubes (*Pélissier et al., 2008*; *Froelich et al., 2011*). GFP demarcated clearly the ER of the PSEs (*Figures 1* and *4*). Unlike HDEL-GFP, the signal from the SEOR-YFP fusion (112 kDa), which is expressed in the cytoplasm of immature SEs, was not restricted to SEs but entered two specific cell files significantly larger than the adjacent PSEs, and did not travel outward beyond this domain (*Figure 4C*). We used resin embedding of the phloem-unloading zone (*Bell et al., 2013*) to localize the YFP signal in transverse sections of the root. The YFP signal was confined to the two large PPP cells that abut the PSE files (*Figure 4D and E*). Time-lapse movies

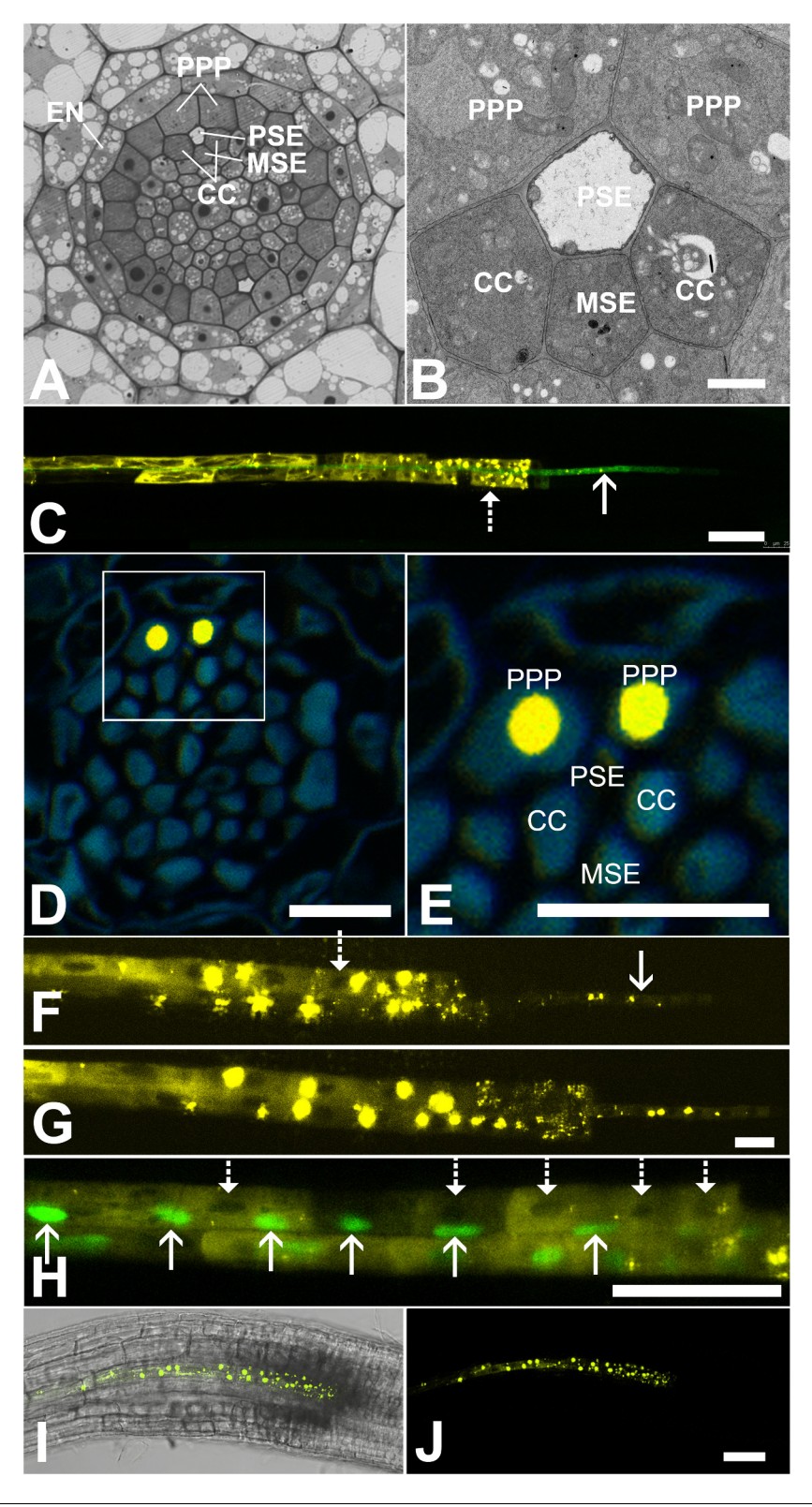

**Figure 4.** Functional organization in the root unloading zone. (**A, B**) TEM images showing a cross section of an Arabidopsis root unloading zone. (**A**) An overview of the central cylinder with phloem pole pericycle cells (PPP), endodermis (EN), companion cells (CC), metaphloem sieve element (MSE), and protophloem sieve element (PSE). (**B**) TEM image of the pentagonal organization of cells surrounding the protophloem file. (**C**) Confocal micrograph of a transgenic Arabidopsis line expressing SEOR-YFP protein (yellow) and GFP targeted to the sieve element ER (green), both under control under a

*Figure 4 continued on next page*

Figure 4 continued

sieve element specific promoter. While GFP is restricted to the ER of the PSE (solid arrow), SEOR-YFP expressed into the cytoplasm escapes into two neighboring cell files (dashed arrow). (D, E) Root cross section of a fixed and embedded transgenic Arabidopsis plant expressing SEOR-YFP protein. The micrograph identifies the two cell files into which SEOR-YFP escapes as the PPP F, (G) Two confocal micrographs extracted from *Video 2* showing SEOR-YFP protein (yellow) in the PPP and PSE. New PPP cells become fluorescent as unloading progresses. Note that small aggregates of SEOR-YFP become increasingly larger basipetal to the unloading zone. (H) Confocal micrograph of a transgenic Arabidopsis line expressing GFP (green) in the nuclei of companion cells (solid arrows) and SEOR-YFP. The nuclei in the CCs do not match the location of the nuclei in the cell files containing SEOR-YFP (dashed arrows), providing further evidence that the two files are the PPP. (I, J) Root tip of a grafted Arabidopsis plant in which the rootstock was wildtype and the scion expressed SEOR-YFP in the shoot. The root was imaged at 10 days after grafting and shows clearly that SEOR-YFP protein has moved from shoot to root, with subsequent unloading into the PPP. Scale Bars; B = 1 µm; F, G = 5 µm; C, D, E, H = 10 µm; I, J = 50 µm.

of the phloem unloading zone supported this observation (*Figure 4F and G*; *Video 2*). Initially, SEOR-YFP was detected only in the immature PSEs as small aggregates. Subsequently, groups of new PPP cells became fluorescent adjacent to the PSEs, presumably when PD connections opened up between the two cell types (*Figure 4F and G*). Using a fluorescent marker for CC nuclei (*Zhang et al., 2008*) we were unable to detect phloem unloading of SEOR-YFP into CCs, only into the PPP (*Figure 4H,I*). This result was unexpected as most models of phloem unloading assume that the exit of macromolecules occurs through CCs (*Patrick, 1997*; *Thorne, 1985*).

It was possible that SEOR-YFP was delivered directly to the PPP from neighboring immature PSEs in the root, rather than by long-distance transport of the protein from the shoot. To test if the latter occurred, we grafted SEOR-YFP scions onto non-transgenic rootstocks. 10 days after grafting, the roots of the grafted plants were indistinguishable from those of the native SEOR-YFP line (*Figure 4I, J*), demonstrating that SEOR-YFP protein, synthesized in SEs of the shoot, is unloaded from PSEs into the PPP.

## Batch unloading of proteins

To monitor the unloading of large proteins in real time, we used the transgenic lines *pAtSUC2-GFP* (27 kDa), *pAtSUC2-ubiquitin-gfp (36 kDa)*, *pAtSUC2-aequorin-gfp* (48 kDa),, and *pAtSEOR-AtSEOR-yfp* (112 kDa; described above) and conducted fluorescence recovery after photobleaching (FRAP) experiments. Time-lapse movies revealed that unloading of large probes (GFP 27 kDa and above) did not occur at a constant rate, as observed with small solutes. Rather, these proteins were suddenly released into the adjoining PPP cells in distinct pulses (*Figure 5*, *Video 3*). We refer to this phenomenon as 'batch unloading', as the proteins are delivered in discrete pulses into individual cells. These cells then became highly fluorescent relative to their neighbors. Batch unloading did not occur at the same time into all cells, but independently into individual cells (*Figure 5*; *Video 3*). It appears that specific domains exist in the root tip that confer a size-dependent filtration of the macromolecules that arrive in the unloading zone. Small proteins such as GFP were batch unloaded and subsequently moved freely throughout the root (*Figure 5A–F*). However, large probes such as aequorin-GFP and SEOR-YFP became trapped inside the PPP after batch unloading (*Figure 5G–J*; *Video 4*).

## Specific induction of callose in the PPP blocks phloem unloading

To further test the role of the PPP in phloem unloading we inhibited the lateral movement of solutes into this cell layer using an inducible callose synthase system (*Vatén et al., 2011*). In this approach a modified, PD-specific, callose synthase (*icals3m*) is induced under estradiol treatment, inhibiting cell-cell movement into the

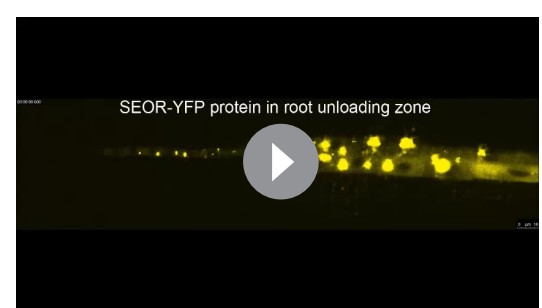

**Video 2.** YFP tagged SEOR-protein in the PSE and two neighboring PPP cell files. During root growth, new PPP cells are integrated into the unloading domain as indicated by tagged protein entering the cells, presumably due to the opening of connecting plasmodesmata.

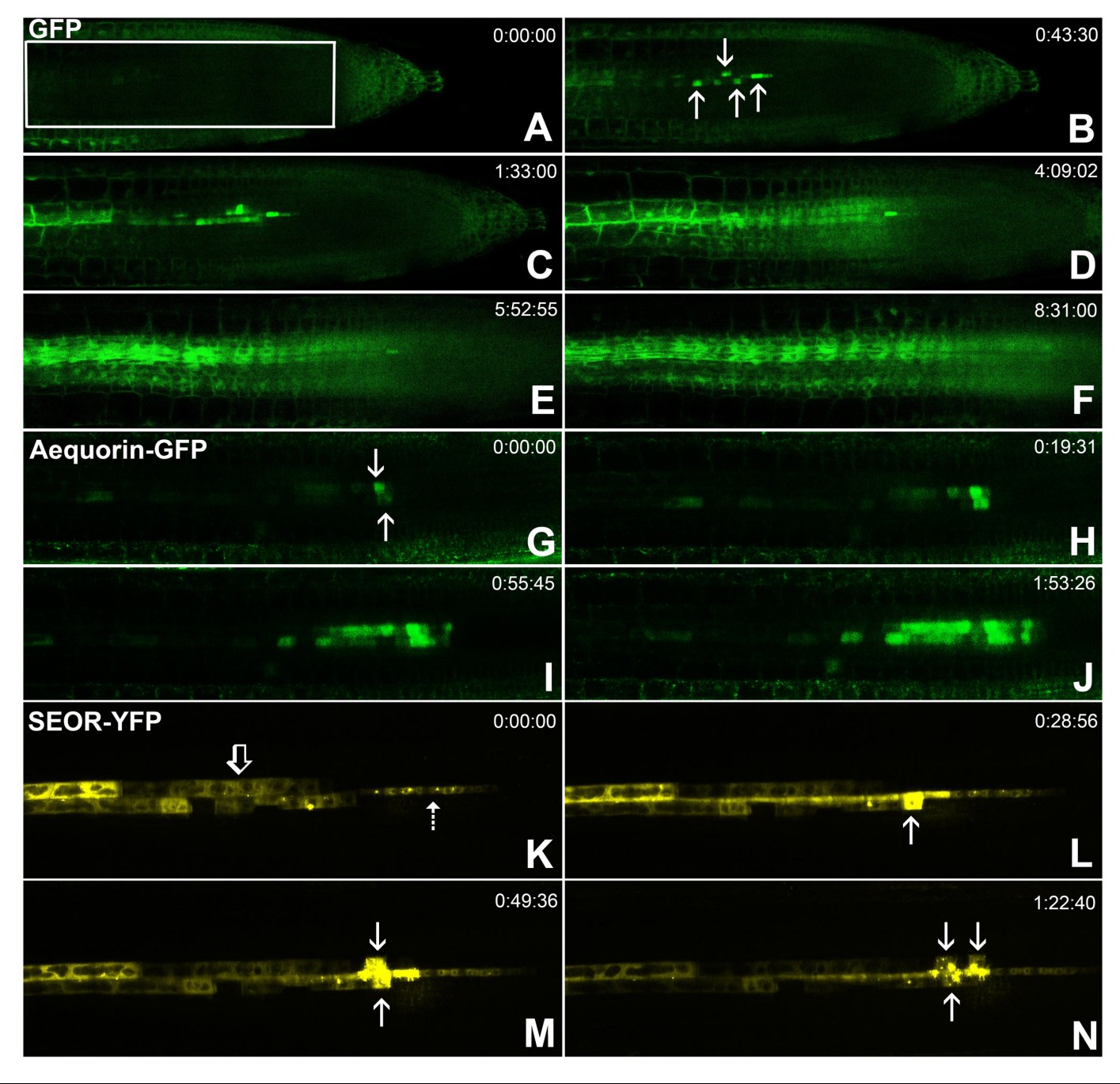

**Figure 5.** Batch unloading of proteins. (A–F) six frames taken from *Video 3*. A) The unloading zone was photobleached (boxed region). Refilling of the unloading zone shows that GFP exits the PSE in discrete batches (arrows in B). Over time, all cells in the root transported GFP until an even distribution of the fluorescent protein was reinstated (C–F). (G–J) Compared to GFP (27 kDa), aequorin-GFP (48 kDa) was batch unloaded but did not move beyond the PPP. (K–N) Four frames extracted from *Video 4* showing batch unloading of SEOR-YFP (112 kDa). In contrast to the CC-expressed GFP probes, SEOR-YFP was expressed in young sieve elements and entered the translocation stream when the sieve-plate pores opened. The immature PSEs are indicated (dashed arrow) and PPP cells are visible (open arrow). When SEOR-YFP aggregates arrive in the phloem unloading zone, they are batch unloaded from the terminal PSEs (solid arrows). As the root continues to extend, the aggregates enlarge and eventually disappear (see also *Figure 4F, G* and *Videos 2* and *4*), probably due to their breakdown in the older PPP cells.

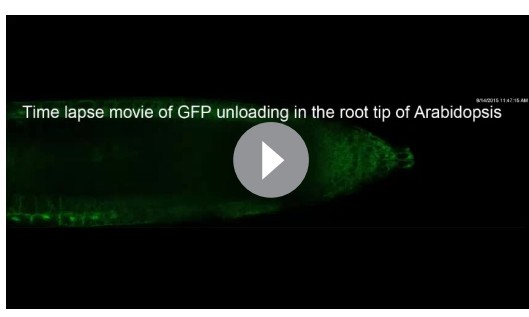

**Video 3.** Time-lapse movie of batch unloading of free GFP (27 kDa).    In situ time lapse movie of a transgenic line constantly supplying GFP into SEs via leakage from CCs where GFP is expressed under control of the SUC2 promoter. After photobleaching of GFP in the unloading zone, refilling reveals that GFP is unloading in batches into individual cells from where it diffuses into the post unloading zone.

induced cell layer of the root (*Vatén et al., 2011*). In Arabidopsis there are twelve CalS proteins, with diverse roles in callose production (*Chen and Kim, 2009*). During the course of this study, we found that CalS8 is expressed specifically in the PPP (*Figure 6A,B*), allowing us to generate an estradiol-inducible line expressing *icals3m* under the CalS8 promoter (*pCALS8::icals3m*). This line was generated specifically to block the connection between PSEs and the PPP. After callose induction we observed a significant arrest in primary root growth relative to roots that were transferred to a non-inducing medium (*Figure 6—figure supplement 1*). Next we labelled the cotyledons of induced and non-induced *pCALS8::icals3m* roots with CF at 8 hr and 24 hr after callose induction. Control roots showed the characteristic unloading pattern of wild-type plants (*Figure 6C*, n = 22) while induced plants showed a severe restriction in phloem unloading. At 8 hr induction, CF was restricted to PSE files with minimal lateral unloading into the root (*Figure 6D*, n = 20). At 24 hr induction, this effect was even stronger. CF did not enter the unloading zone and remained completely restricted to PSEs distal to the unloading zone (*Figure 6E*, n = 9). To determine the potential role played by CCs in phloem unloading, we used the *sister of apple* (*sAPL*) promoter to drive the production of the mutant CALS3 protein (*psAPL::icals3m*). We chose this promoter as other CC-specific promoters (e.g. *SUC2*) were expressed only weakly within the phloem-unloading zone (data not shown). Within the phloem unloading zone the psAPL promoter was expressed strongly in CCs and MSE, but not in the PPP (*Figure 6F,G*). As above, CF transport in these seedlings was monitored at 8 hr and 24 hr after estradiol induction. At both time points CF was unloaded without restriction, and induced and non-induced seedlings showed an identical pattern of unloading (*Figure 6H–J*, n = 3, n = 3, n = 2, respectively). Furthermore, we found that callose induction in the *psAPL::icals3m* seedlings did not significantly affect root growth relative to the uninduced control plants (*Figure 6—figure supplement 1*). To confirm that callose synthase was overexpressed in the predicted cell layers, roots were fixed, embedded and stained for callose after the CF transport assays. Control roots showed the characteristic callose labelling associated with PSE files (*Figure 6K*; see also *Figure 3*). Confocal imaging of *pCALS8::icals3m* seedlings confirmed that callose was deposited strongly in the PPP relative to other cells surrounding the PSE files (*Figure 6L–O*), while *psAPL::icals3m* roots showed strong callose labelling in the CC files, as predicted (*Figure 6—figure supplement 6P–S*). Collectively, these data provide strong evidence that phloem unloading occurs predominantly via the PPP, not CCs, and suggest that the PD between PSE and PPP are central players in the unloading process.

## A PD-based molecular filter regulates phloem unloading

In order to understand the structural basis for the differential filtration of molecules in the unloading zone, we investigated the ultrastructure of PD at all cell interfaces between the protophloem file and its five interconnecting cells (See *Figure 4A and B*). We found different PD types in the unloading zone (*Figure 7A*). PD between the PSE and MP were simple, or occasionally branched (*Figure 7B*). Between PSEs

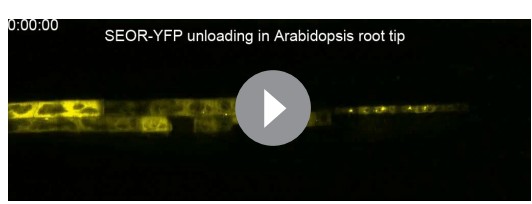

**Video 4.** Batch unloading of YFP tagged SEOR protein (112 kDa).    SEOR-YFP protein is expressed in young sieve elements and remains as aggregates in the sieve elements after degradation of the nucleus. The time-lapse movie shows batch unloading of this large protein into the PPP.

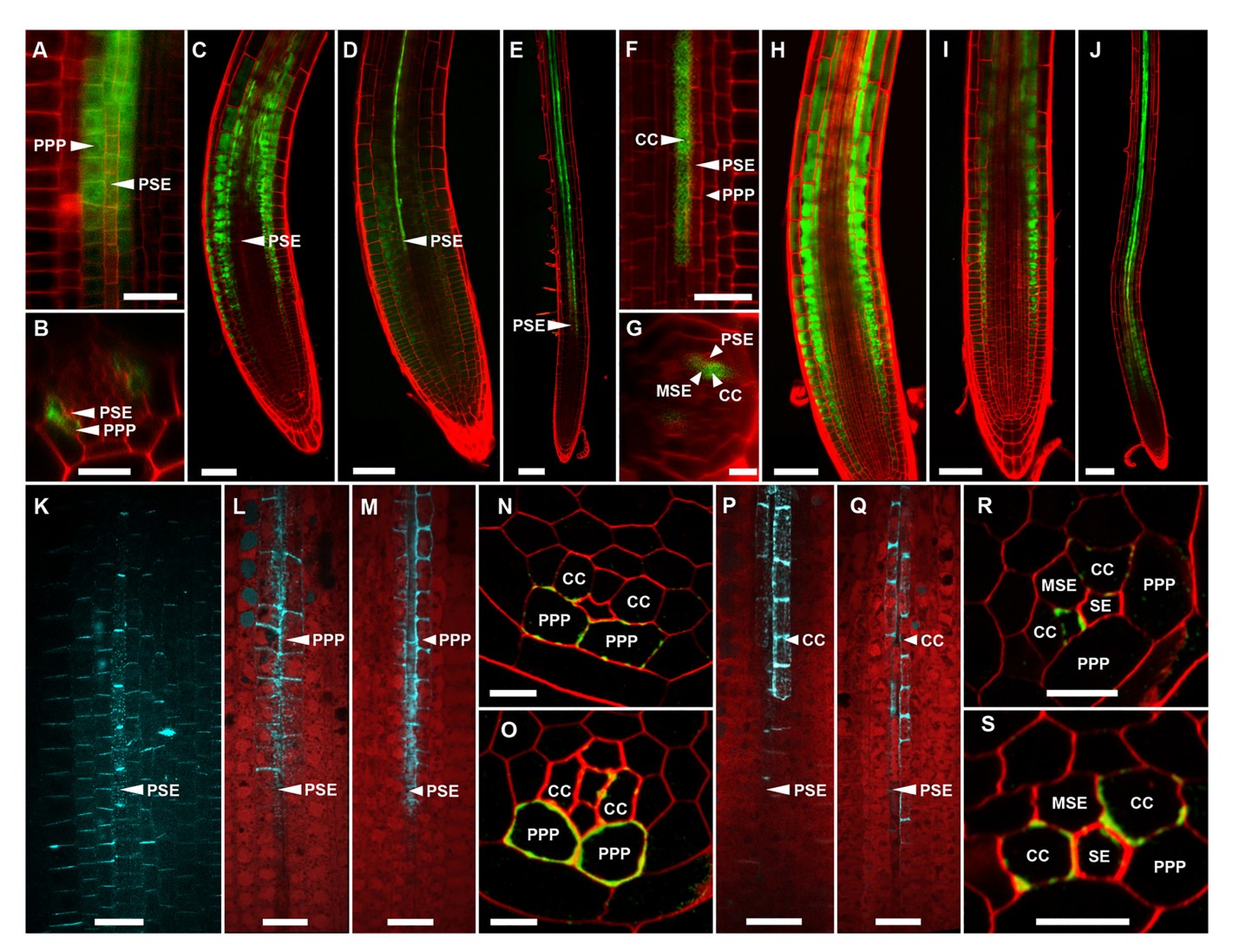

**Figure 6.** Callose induction in the PPP, but not CCs, blocks phloem unloading. (A) *pCALS8::ER-YFP* is expressed exclusively in the PPP. (B) Transverse optical section of A. (C) CF unloading in a control root expressing *pCALS8::icals3m* transferred to non-inducing medium. Unloading progresses as in wild-type roots. (D) CF unloading is restricted to the PSE files in *pCALS8::icals3m* roots at 8 hr after callose induction in the PPP. (E) As D but at 24 hr post-callose induction in the PPP. (F) psAPL promoter expression (psAPL-GFP) is restricted to CCs and MSE. (G) Transverse optical section of F. (H) CF unloading in a control root expressing *psAPL::icals3m* transferred to non-inducing medium. (I) CF unloading is not restricted in *psAPL::icals3m* roots at 8 hr after callose induction in CCs. (J) As I but at 24 hr post-callose induction in CCs. (K) Sirofluor staining of a control root showing general background staining of PD around PSE files. L) Sirofluor staining of a *pCALS8::icals3m* root at 8 hr after callose induction in the PPP. (M) Sirofluor staining of a *pCALS8::icals3m* root at 24 hr after callose induction. In both L and M the roots were stained immediately after CF transport. (N) Callose immunolabelling (green) of a *pCALS8::icals3m* root at 8 hr after callose induction in the PPP. Cell walls are labelled red. (O) As N but at 24 hr after callose induction. (In addition to the PPP, sometimes callose staining is also observed in the CC). (P) Sirofluor staining of a *psAPL::icals3m* root at 8 hr after callose induction in CCs. (Q) As P but at 24 hr after callose induction. (R) Callose immunolabelling (green) of a *psAPL::icals3m* root at 8 hr after callose induction in CCs. (In addition to the CC, callose staining was sometimes observed in the MSE). Cell walls were counterstained with calcofluor white (labelled red). (S) As R but at 24 hr after callose induction. Scale bars: N, O, R, S: 5 um. A, B, F, G, K, L, M, P, Q: 10 um. C, D, H, I, J: 50 um. E, J: 100 um. Abbreviations as in *Figure 4*.

The following figure supplement is available for figure 6:

**Figure supplement 1.** Growth of *pCALS8::icals3m* and *psAPL::icals3m* seedlings on 5 uM beta estradiol relative to uninduced controls (mock DMSO).

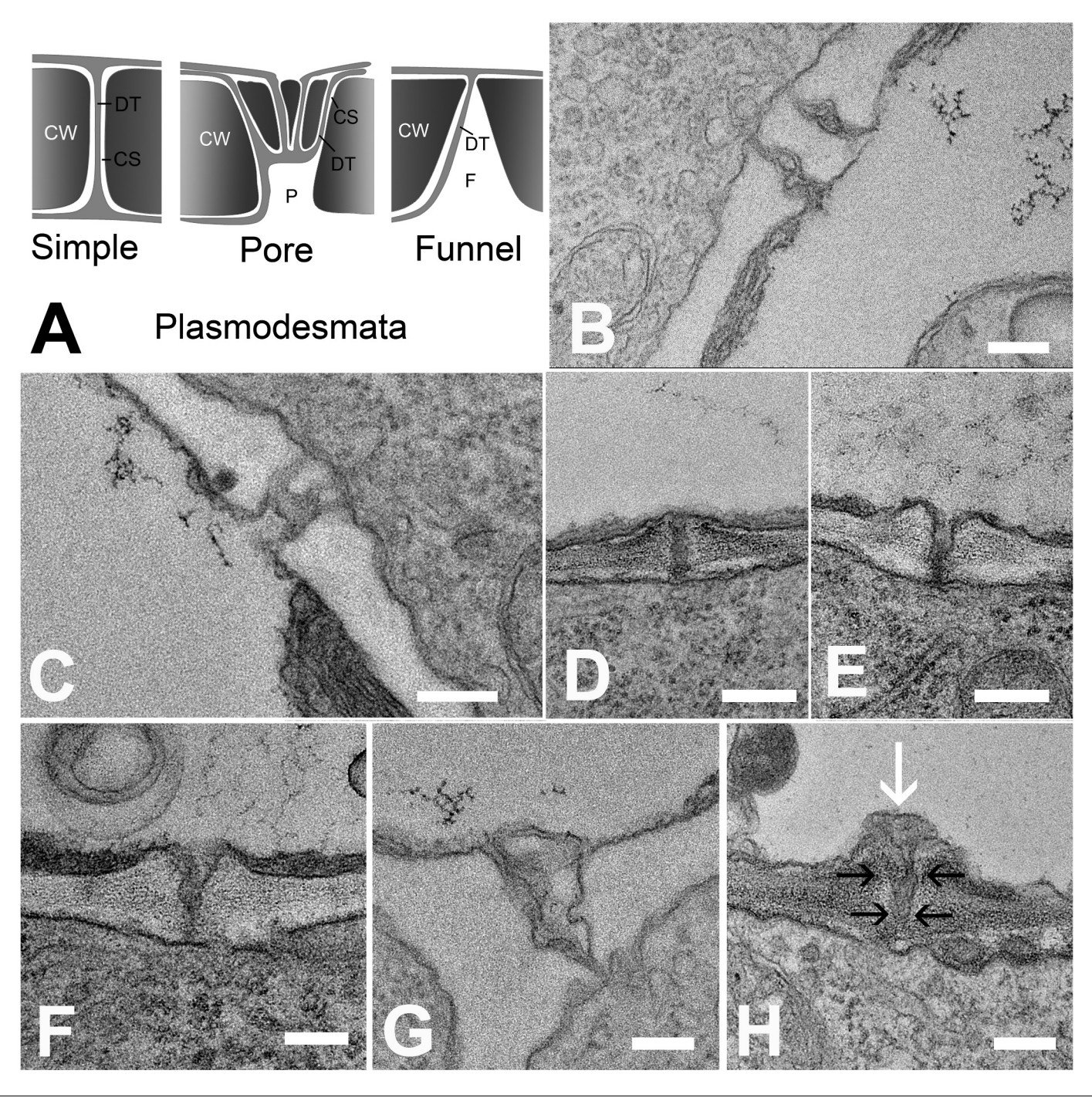

**Figure 7.** Types of plasmodesmata connecting different cell interfaces. (**A**) Schematic diagrams of the different plasmodesmata connecting protophloem sieve elements to surrounding cell types. (**B**) Image of the PSE-MSE interface showing a cell wall with two simple plasmodesmata. (**C**) A pore-plasmodesma in the cell wall between PSE and CC. (**D–I**) Plasmodesmata connecting PSE with PPP. (**D**) Simple plasmodesmata, found rarely. (**E–H**) Funnel plasmodesmata. These showed a wide opening on the PSE entrance tapering towards the PPP. (**H**) Electron-dense components (white arrow) of unknown composition were often observed within funnel plasmodesmata (black arrows). DT = desmotubule, CW = cell wall, P = pore, F = funnel, CS = cytoplasmic sleeve. Scale bars: B, C, G, H = 200 nm; D, E, F = 500 nm.

and CCs, PD displayed the typical single pore on the PSE side and multiple branches towards the CC (*Figure 7C*), as shown often for connections between SEs and CCs in other tissues (*Esau and Thorsch, 1985*; *Oparka and Turgeon, 1999*). The structure of most of the PD between PSE and PPP, however, was quite different and showed an architecture not previously described. In addition to a few simple PD (10%; n = 20), funnel-shaped PD with apertures of up to 300 nm diameter were found at the PSE entrance, tapering towards the PPP entrance (*Figure 7D–H*).

In order to extract the number of PD available for unloading, we conducted serial block-face scanning electron microscopy followed by 3D reconstruction of the interface between PSEs and the adjacent cells (*Denk and Horstmann, 2004*; *Furuta et al., 2014*; *Video 5*). Using this method, we collected 2100 serial transverse sections that spanned six complete SEs (*Figure 8A*), including the junction between PSE zero and the adjacent immature PSE (*Figure 8B*). During reconstruction, we attributed the exact position of each PD to the different wall interfaces shared by the PSE, color-coded to ease identification (*Video 5*). Using 3D reconstruction, we could image PD on both the outer (*Figure 8C*) and inner (*Figure 8D*) walls of the PSE files. A movie showing a 'fly through' of the interior of the PSE file is shown in *Video 5*. PD relative distribution was 45.3%, 40.8%, and 13.9% between PSE-PPPs, PSE-CCs, and PSE-MSE, respectively, when corrected for the total interface between the cell types. Thus, PD leading from the PSE to PPP and CCs are equally abundant while those to the MSE are significantly lower. Next, we acquired data for the total numbers of PD along the entire phloem-unloading zone. We found that the total number averaged 527 ± 58 (n = 4) per protophloem unloading domain (since Arabidopsis roots are diarch, twice the number is available for unloading within the entire root tip). Based on the percentage distributions of PD obtained using serial sectioning, we calculated that in the phloem-unloading zone of the root there are approximately 240 plasmodesmata at the PSE-PPP interface, 215 plasmodesmata at the PSE-CC interface, and 73 plasmodesmata at the PSE-MP interface potentially available for unloading.

## A new model of phloem unloading in Arabidopsis roots

Current models of phloem unloading, such as the high-pressure-manifold model (*Fisher, 2000*; *Patrick, 2013*) assume high pressure differentials between PSEs and the unloading zone in order to drive fluids and solutes through the PD into the neighboring cells. Recently, direct sieve tube pressure, viscosity, flow velocity, and tube geometry measurements in morning glory showed, however, that the majority of energy provided by the pressure differential between source and sink is consumed to drive flow, and that a high pressure differential in the root system is unlikely to exist, calling the current model into question (*Knoblauch et al., 2016*). In principle, there are two possibilities of how the phloem sap might escape the PSEs. Bulk flow could move the entire volume of solvent and solutes through the PD into neighboring cells. Alternatively, the solutes could diffuse through PD while the solvent (water) could be removed from PSEs via membrane leakage, potentially via aquaporins (*Doering-Saad et al., 2002*). Finally, a combination of diffusion and bulk flow could account for the observed transport.

In order to evaluate the feasibility of the different routes, we gathered the necessary parameters to model flow in the phloem-unloading zone (*Table 1*). We measured the average phloem flow velocity in the protophloem translocation zone to be $u = 22.6 \pm 5.1$ µm/s (n = 11) and the average sieve tube diameter to be: $d = 3.6 \pm 0.44$ µm (n = 11) resulting in a volume flow rate of $Q = \frac{\pi}{4}d^2u = 230$ femtoliter/s (230 µm³/s) to be unloaded from a single PSE file. Literature values on phloem sucrose concentration average about 500 mM (*Hall and Baker, 1972*; *Kallarackal et al., 1989*; *Mendoza-Cózatl et al., 2008*; *Winter et al., 1992*), hence the total amount of sugar to be unloaded is approximately $I = Qc = 1.2 \times 10^{-13}$ mol/s. The solutes must leave the phloem through the PD,

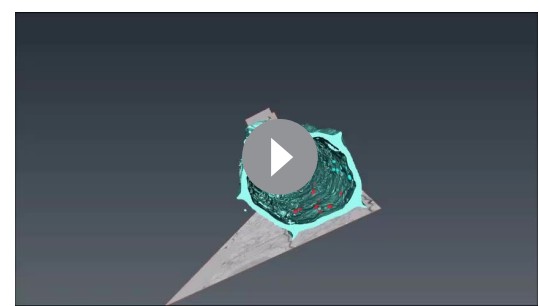

**Video 5.** 3D volume reconstruction of serial block face data. The movie shows the volume reconstruction of the PSE file with highlighted cell walls. Color coding reveals the location of plasmodesmata connecting the PSE to the neighboring cell types.

while some of the solvent (water) may be removed via membrane leakage. To facilitate transport, an average of 240 PD are available at the PSE/PPP interface, while the water may leak anywhere along the 350 µm-length of the unloading zone. For simple PD we assumed a cylinder of constant dimensions with a desmotubule of 15 nm diameter and a cytoplasmic sleeve width of 2.8 nm, corresponding to the hydrodynamic radius of GFP. Based on our observations that GFP is batch unloaded, but diffuses relatively freely throughout the root, this molecule appears to be slightly larger than the size exclusion limit imposed by the neck region of funnel PD. For funnel PD we assumed the same dimensions on the PPP side but an average funnel opening of 150 nm (*Figure 7*) towards the PSE. The location of the desmotubule in funnel PDs impacts the resistance which led us to assume two extremes which are discussed in detail in the appendix. The cell wall thickness (length of the simple and funnel PDs) was taken to be 500 nm.

To elucidate the role of funnel PD in unloading, we first considered the conditions necessary to facilitate unloading by simple PD. We found that the required pressure differential to drive unloading solely by bulk flow through simple PD would be 8.14 MPa (see Appendix for detailed calculations). This pressure differential has neither been measured in SEs, nor can it be considered as feasible. However, assuming funnel PD instead of simple PD, the required pressure would be as low as 0.05–0.2 MPa. Such a relatively low pressure differential would be facile to maintain between plant cells. Because the viscous resistance in wide pores is greatly reduced (Appendix), funnel PD are much more efficient in unloading compared to simple PD.

To estimate the solute concentration difference that would be required to account for unloading by pure diffusion, we assumed a length of the unloading zone of 350 µm. For the geometrical and physiological values listed above, a concentration difference of 276 mM would be required for diffusive unloading through funnel PDs. In this case, however, a second route for unloading of the solvent (water) would be necessary. The permeability of the plasma membrane varies significantly between cell types and values of $10^{-14}$ to $10^{-12}$ m/s/Pa have been reported (*Kramer and Boyer, 1995a*). Considering the parameters outlined above, a pressure of 0.059 to 5.9 MPa would be needed to remove the solvent from the PSE. Thus, at a high membrane permeability removal of the solvent over the membrane would be feasible.

Our analysis of unloading kinetics leads us to conclude that the distinctive funnel shape of the SE-to-PPP PD is crucial to enabling efficient unloading. In this case, convective phloem unloading, i.e. a combination of diffusion and bulk flow, is feasible, and that relatively moderate pressure and concentration differentials are necessary to drive transport. In our calculations (Appendix) we have assumed that PD connecting PSEs with CCs and MSEs play little or no direct role in the unloading process, consistent with our results using the *icals3m* system (*Figure 6*). In this scenario, the unique architecture of funnel PD could still accommodate the calculated unloading rates (Appendix)

## Discussion

### Convective phloem unloading

Based on our structural and physiological data bulk flow is likely to be the dominant mechanism of unloading. Bulk flow would require a pressure differential of about 0.05–0.2 MPa, equal to an osmotic potential difference of about 20–80 mM. Solute unloading by diffusion alone, on the other hand, would require a concentration difference of 276 mM. While bulk flow predominates, the above mechanisms are not exclusive, and both may contribute to unloading. Bulk flow requires a pressure differential generated by osmosis which would lead to a concentration difference between PSEs and PPPs. This would induce diffusion, in parallel with bulk flow, even if its contribution is lower. The term for the combination of diffusion and bulk flow is convection, which leads us to propose a new model of Convective Phloem Unloading.

### Post-phloem transport and size-dependent filtration

We have identified different domains within the root tip that provide a size dependent filtration of molecules. Cross sections of Arabidopsis roots (*Figure 9A*) show the typical diarch structure. The pentagonal architecture of the cell complex surrounding the PSEs and the different PD connecting the cell types is depicted in *Figure 9B and C*. Our results indicate that the PSE-PPP interface is the principal route for all solutes to be unloaded. Diffusion will, however, lead to a relatively quick

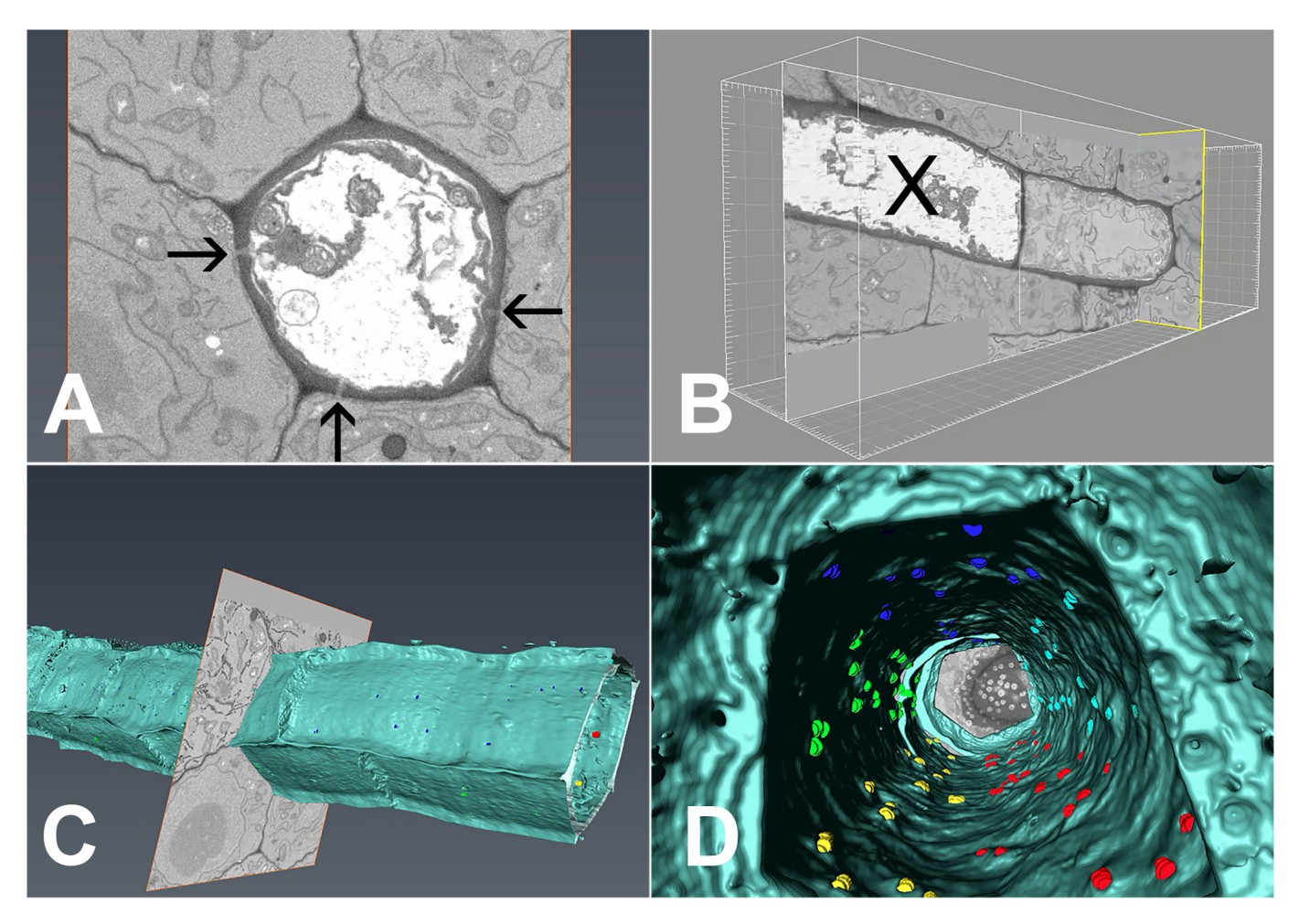

**Figure 8.** 3D overview of protophloem sieve elements in the root tip obtained by serial block-face scanning electron microscopy. (**A**) Cross-section of one phloem pole in the unloading zone. PD are indicated by darts. (**B**) Longitudinal section of the protophloem unloading zone. PSE zero (**X**) is connected to a neighboring immature protophloem sieve element. (**C, D**) 3D longitudinal view of the protophloem unloading zone. Serial sections were used to reconstruct the unloading zone and quantify PD connections from PSEs to adjacent cells. (**C**) shows PD on the outer face of the PSE. (**D**) is derived from *Video 4* and shows the PD on the inner faces of the PSE. In the images, PD are color coded (blue/cyan PSE-PPP, red/green PSE-CC, and yellow PSE-MSE).

redistribution of the solutes through simple PD connecting the cells within the post-phloem unloading zone, along a solute concentration gradient (*Figure 9D,E*). This gradient will also direct solutes to areas of highest demand, while higher consumption will lead to steeper gradients.

Unloading of small proteins such as GFP (27 kDa) occurs by batch unloading through funnel PD into PPPs. Once unloaded, GFP enters the post unloading zone and is evenly distributed throughout the root tip (*Stadler et al., 2005*). Large proteins of the size of aequorin-GFP (48 kDa) or SEOR-YFP (112 kDa), however, remain trapped within PPPs and cannot enter the post-phloem pathway to the meristem. Clearly, the different types of PD between PSEs and neighboring cells have major functional impacts on molecular flow in the root tip.

Many proteins have been found in extracted phloem sap, most showing no obvious function in long-distance signaling (*Turnbull and Lopez-Cobollo, 2013*; *Batailler et al., 2012*; *Paultre, et al., 2016*). Thus, the question arises as to how many of these molecules are components of the phloem sap by default rather than by design (*Paultre, et al., 2016*). To address this point we need to consider where such macromolecules originate. Many systemic macromolecules are thought to arise in CCs and pass into SEs through the pore-PD that connect SEs with CCs (*Fisher et al., 1992*;

**Table 1.** Base parameters used to model phloem unloading.

| | Assuming transport through simple PD at PSE/PPP interface | Transport through PD at PSE/PPP interface |
|---|---|---|
| Length of unloading zone | 350 µm | 350 µm |
| Desmotubule Diameter | 15 nm | 15 nm |
| Cytoplasmic Sleeve Diameter | 2.8 nm | 2.8 nm |
| Cell Wall Thickness | 500 nm | 500 nm |
| Phloem Sap Osmotic Potential | 500 mM | 500 mM |
| Funnel opening towards PSE | – | 150 nm |
| # of PD available for Unloading | 240 simple PD | 24 simple PD, 216 funnel PD |
| Total Sap Volume | 230 fl/s | 230 fl/s |
| Required Pressure Differential | 8.14 MPa | 0.05–0.2 MPa |

*Lucas et al., 1996*; *Sjolund, 1997*; *Oparka and Turgeon, 1999*). There appears to be selectivity to this movement, large proteins (above 70 kDa) are restricted from entering the translocation stream while those below this cutoff are not (*Paultre, et al., 2016*). Interestingly, in our current study SEOR-YFP (112 kDa), which is translated in immature SEs rather than CCs, was able to unload into the PPP, suggesting that the size exclusion limit that regulates macromolecular exchange between PPP and SE in sinks may be larger than between SEs and CCs in source phloem tissues. Another potential origin of macromolecules in phloem sap is young developing sieve elements. As shown in *Figure 1*, PSEs start to conduct when their nucleus degenerates. At this point the cytoplasmic content in the PSEs is still very dense. After integration into the mature PSE file, there is no other route left for degraded structures but phloem unloading, most likely into the PPP. Unlike the xylem pole pericycle (XPP), which is involved in lateral-root formation (*Parizot et al., 2012*) and apoplastic xylem loading (*Takano et al., 2002*), a specific role for the PPP has not been proposed, although its transcriptome shares similarity to the underlying protophloem (*Parizot et al., 2012*). We have now identified the 'post-phloem domain' previously described by *Stadler et al. (2005)* and *Paultre, et al., 2016* as the PPP. It appears that while the XPP may be involved in apoplastic xylem loading, the PPP is intimately involved in symplastic phloem unloading, and may function as a repository for degraded PSE contents as well as systemic macromolecules. Although PD connect all cell types surrounding the PSEs, our data suggest that exit via the PPP is the major route of unloading in Arabidopsis roots. In mature source tissues, as well as in secondary sieve elements produced by the cambium, macromolecules will arrive in the phloem-unloading zone. This was demonstrated when we grafted SEOR-YFP scions onto wild-type rootstocks (*Figure 4I,J*). Without continuous macromolecule unloading, PSEs would fill up rapidly and phloem unloading would be impaired (*Paultre, et al., 2016*). PPPs are the major recipients of systemic macromolecules and it might be expected that they are specialized for the degradation of proteins and RNAs, an interesting area for future research. PPPs may also protect the downstream cells from receiving unwanted, potentially interfering mRNA molecules.

How does the root distinguish between true signaling molecules and those moving by default in the phloem? One possibility is that unloaded macromolecules destined to traffic further than the PPP may have specific sequences that interact with the PD that connect the PPP to the endodermis and beyond. Phloem-mobile viruses are able to move beyond the PPP suggesting that they can break the 'barrier' normally imposed by the PPP (*Valentine et al., 2004*). A clear future challenge will be to identify systemic macromolecules that can move beyond the PPP, and to analyze these macromolecules for specific motifs that, like transcription factors (*Xu et al., 2011*), allow them to interact with and modify PD.

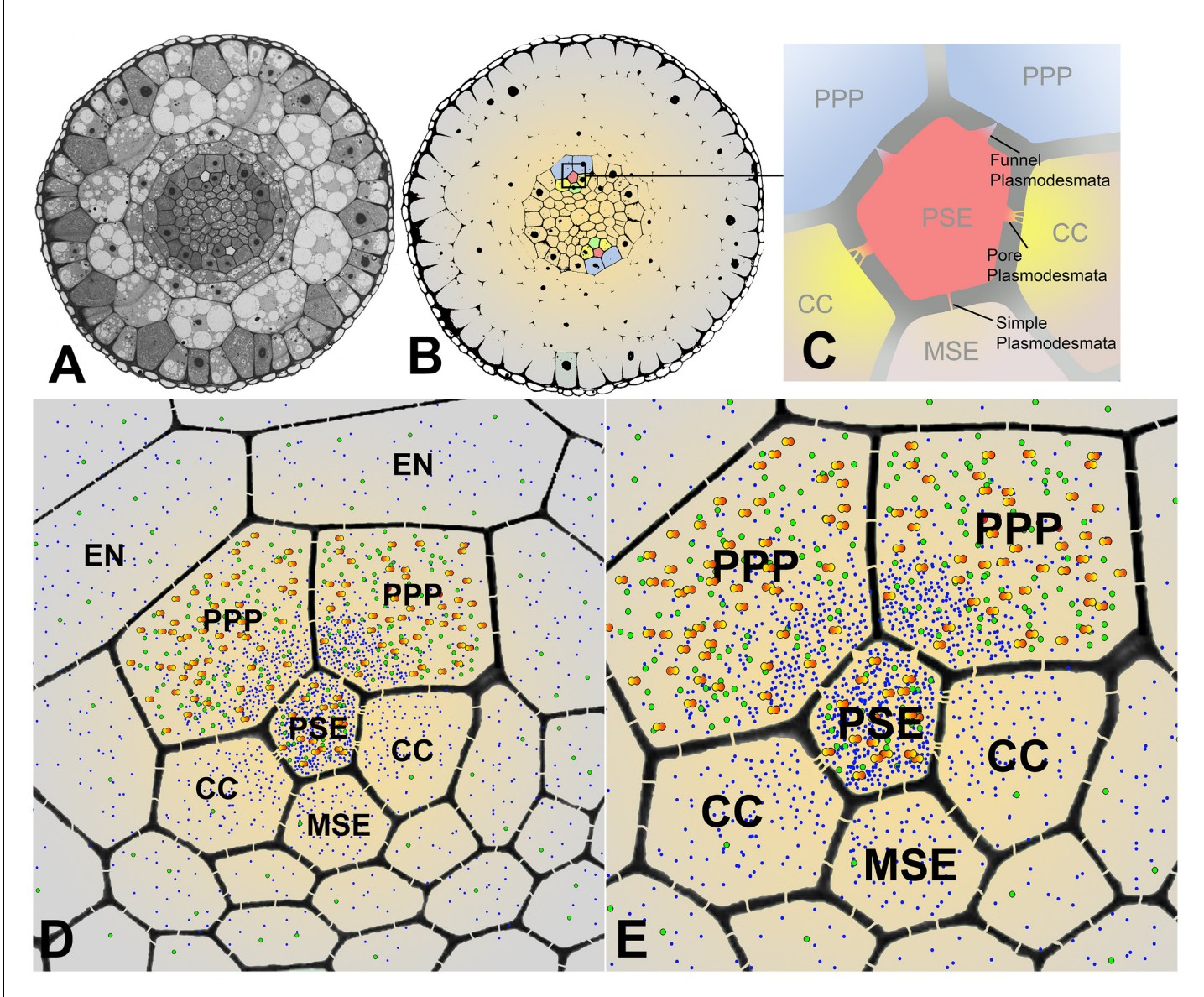

**Figure 9.** Cross sectional overviews of the Arabidopsis roots showing PD connections and size-dependent phloem unloading of solutes and macromolecules. (**A**) Standard light micrograph showing a cross section of the Arabidopsis root. (**B**) A false colored cross section of the Arabidopsis root highlighting the two phloem poles in the unloading zone. (**C**) Diagram of the cells in the phloem pole and the types of PD that connect the PSEs to each adjacent cell. PPP cells are connected to the PSE by funnel PD, CCs are connected by pore-PD, and MSEs are connected by simple PD. (**D and E**) Diagram showing the location of various solutes and macromolecules depending on molecular mass. Once unloaded via the PPP, sucrose (blue dots) and GFP (green dots) are able subsequently to enter all cell types via PD. However, larger macromolecules such as SEOR-YFP (yellow/orange dots) are unloaded only into PPP cells. EN = endodermis.

## Materials and methods

### Plant material

*Arabidopsis thaliana* were grown in microROCs (Advanced Science Tools LLC. Pullman, WA) as described elsewhere (*Froelich et al., 2011*). Soil was kept saturated by placing the microROCs in 1–2 inches of water daily for 5 min creating a soil water saturation of >80%. Chamber conditions were 16 hr photoperiods with 100–200 µE m$^{-2}$ s$^{-1}$ at 18°C to 22°C.

## Fluorescence recovery after photobleaching

Experiments were conducted on a Leica SP8 confocal microscope equipped with a supercontinuum laser (470–630 nm) and a pulsed 405 nm diode laser. 5(6)-Carboxyfluorescein diacetate (CFDA) stock solution, 5 mg/ml in acetone, was diluted 1:10 (v/v) in ddH$_2$O and loaded into the phloem through application to cut cotyledons at 7–10 days post-germination as described earlier (*Knoblauch et al., 2015*). Sieve elements in the most apical root protophloem loaded with CFDA were photo-bleached with 480 nm, 488 nm, and 496 nm lasers concurrently at maximum power and 6.5x zoom with a 20x lens. Bleaching started apically and moved toward the hypocotyl until the entirety of CFDA in the sieve tube in the field of view was bleached. Subsequent recording of the refilling occurred immediately following photo-bleaching at 200 Hz and 0.75x digital zoom by excitation with the 488 nm laser line at 15% of continuous power. Emission was collected between 505 and 545 nm.

## Arabidopsis grafting

Seeds of Arabidopsis wt and the transgenic line pAtSEOR-AtSEOR-yfp were sterilized with 8% bleach and 1% TWEEN-20. After 5x wash in distilled water seeds were plated on MS with 1.2% agar and 0.2% sucrose, pH 5.7. After 48 hr stratification in the dark at 4°C seeds were oriented vertically at 23°C with 18 hr photoperiod. After 5–7 days, seedlings were grafted following the hypocotyl-grafting procedure of *Turnbull et al. (2002)* consisting of a transverse cut and butt alignment with silicon collars. The seedlings were cut transversely in the upper region of the hypocotyl with ultrafine microknives (Interfocus, n°10315–12). Scions were grafted onto wild type stocks using a short silicon collar for support on MS agar plates. The grafts were left to grow under LD with the plates still oriented vertically until new lateral roots of the stocks were fully established (~10 days). The grafts were imaged between 5 and 10 dag.

## Fixation and embedding of fluorescent proteins

Ten days old seedlings of transgenic *Arabidopsis thaliana*, expressing AtSEO::SEO-YFP, were embedded in LR White following the method described by *Bell et al. (2011)*. Tissue samples were fixed overnight at 4°C in a solution of 4% formaldehyde, 1% glutaraldehyde, 50 mM 1,4-piperazine-diethanesulofonic acid (PIPES) and 1 mM CaCl$_2$. The samples were then washed in buffer (50 mM PIPES, 1 mM CaCl$_2$) three times for 10 min before dehydration in a graded ethanol series (50%, 70%, 2 × 90%). The tissue samples were then infiltrated at 4°C in medium grade LR White at 1:1, 1:2, 1:3, 1:9 ratios of 90% ethanol:resin for 45 min each before two 60 min changes in 100% LR White. The final embedding step was done at ambient temperature. The samples were then polymerised in gelatin capsules (TAAB) at 50°C for 24 hr.

## Confocal imaging

Esculin and Sirofluor were excited with a pulsed 405 nm diode laser. Emission was collected with a hybrid detector between 420 and 480 nm or 420 and 600 nm respectively. GFP and esculin were imaged with sequential scan for fluorescence emission separation. For GFP, excitation was 488 nm and emission detection with a hybrid detector at 495–535 nm. Esculin excitation was 405 nm and emission collection with a hybrid detector at 420–470 nm. GFP and YFP were sequentially scanned for separation of emission fluorescence. GFP excitation was 484 nm at 50% continuous power. Emission collected with a hybrid detector at 489–505 nm. YFP excitation was 514 nm at 15% continuous power. Emission collected at 519–564 nm with a hybrid detector.

## Transmission electron microscopy

Samples were chemically fixed in the microwave with 2% glutaraldehye, 2% paraformaldehyde and 2% DMSO in 0.1 M cacodylate buffer. Microwave fixation time was 2 min, followed by a 2 min break and a final 2 min microwave period at maximum temperature of 25°C. Samples were then washed 5x for 10 min in 0.1 M cacodylate buffer. Postfixation took place in 1% osmium tetroxide (0.1 M cacodylate) overnight at 4°C. Samples were rinsed 3x for 10 min. in 0.1 M cacodylate buffer. Dehydration was carried out with methanol in the microwave for one minute followed by 5 min at room temperature for each step in 5% increments from 5% to 100% methanol. 100% methanol was replaced twice and the samples were then transferred to 100% propylene oxide with three exchanges. Samples were embedded in Spurr resin at the following propylene oxide:Spurr ratios: 2:1 for 2 hr, 1:1 for 2

hr, 1:2 for 2 hr, and 100% Spurr overnight 3X. The resin was cured at 65°C for 24 hr. Samples were sectioned and stained with 1% uranyl acetate for 6 min followed by a ddH$_2$O rinse and stained in Reynold's lead for 6 min. Following Reynold's lead samples were quickly rinsed in 0.1 N NaOH and ddH$_2$O. Samples were imaged with an FEI T20 at 200kv.

## Serial block face scanning electron microscopy (SBFSEM)

Material for SBFSM was prepared as described by Furuta et al. (2014). Wild-type Columbia roots were fixed with 2.5% glutaraldehyde, 2% formaldehyde in 0.1 M Na-Cacodylate buffer (pH 7.4) supplemented with 2 mM CaCl2 for 2–3 hr at room temperature and embedded in Durcupan ACM resin (Fluka, Sigma-Aldrich). After standard dehydration steps, samples were embedded in silicone holders filled with 100% Durcupan and infiltrated for at least 2 hr before polymerization at 60°C. The roots were trimmed to the desired starting point from the tip using an EM Ultracut UC6i ultramicrotome (Leica Mikrosysteme GmbH). Images were acquired with a FEG-SEM Quanta 250 (FEI, Hillsboro, OR), using a backscattered electron detector (Gatan Inc.). The block faces were sectioned at 40 nm increments. The images were initially processed and segmented using Microscopy Image Browser, a self-developed program written under Matlab environment (Mathworks Inc.) and available at http://mib.helsinki.fi/. A SBFSEM dataset of 2100 images was subsequently segmented and visualized in 3D using Amira 6.0 software (FEI Corp). The outer structure cell walls were discriminated in a hybrid method of interpolated interactive masking and grey-level segmentation. The individual plasmodesmata were identified and segmented sequentially by a manual interactive method and color-coded into groups that related to the particular cell wall interface along the protophloem files.

## Construction of inducible lines

The promoters of CALS8 (At3g14570) and sAPL (At3g12730) were amplified using the primers CALS8-AscI-F AAGGCGCGCCCGGCAACATGAAATACGGGA and CALS8-XhoI-R ACAGCTCGAGG TTTTGGGAGAAAATCAATCAGAA, and SAPL-AscI-F AAGGCGCGCCAGCTAATAAGAAAGGGAGA TCTCTG and SAPL-XhoI-R ACAGCTCGAGTTAACTAACAAAGTACTAAATGCCGA, respectively) and cloned into P4P1RpGEMt containing the estrogen receptor XVE (Zuo et al. 2000). Using the Multi-Site Gateway system (Invitrogen), the inducible promoters were combined with the *icals3m* construct and the nopaline synthase terminator in destination vector pBm43GW. Arabidopsis Col-0 plants were dipped with the different constructs and positive transformants were selected using Basta (*pCALS8*) and Hygromycin (*psAPL*). Lines with single insertions were selected in T2 and homozygous plants were obtained in T3.

## Growth assays and phloem unloading analyses

*pCALS8::icals3m* and *psAPL::icals3m* seeds were stratified for 2 days at 4°C in the dark and were sown on media containing 0.5 MS and 1.2% plant agar, pH 5.8. At 4 days post sowing, seedlings were induced in plates containing the same media plus 5 µM Beta estradiol or the corresponding amount of DMSO (mock). Plates were scanned at times 0, 3, 8 and 24 hr after the induction. Roots were measured using ImageJ and seedlings were subsequently used for CFDA loading. 1 mM CFDA in a solution of 72% acetone in water was mixed with 0.5% Adigor solution (Syngenta) to 10:1 proportions. 0.2 µL of this solution was applied to the adaxial side of the leaves, separating the leaves from the media to avoid undesired dye spreading. Approximately 10 roots were used for each treatment. 20 min after loading, roots were mounted in propidium iodide and imaged by confocal laser scanning microscopy (Zeiss LSM700).

## Callose detection

Seedlings used for phloem unloading analysis subsequently underwent callose detection experiments through Sirofluor staining and callose immunolocalization allowing three independent experiments on the same biological material. Callose accumulation on the CFDA positive roots used for unloading experiments was confirmed first by Sirofluor staining. Whole seedlings were incubated 2 hr at room temperature in a 25 µg/mL (42.66 µM) solution of Sirofluor fluorochrome (Biosupplies, AU) in a 50 mM K$_2$HPO$_4$ buffer under vacuum (60 mPa). Roots were then dissected and mounted in a 1:1 solution of AF1 antifadent (Citifluor) and 67 mM K$_2$HPO$_4$ buffer, and subsequently imaged by

confocal laser scanning microscopy (Zeiss LSM700). Following Sirofluor staining, roots were fixed for subsequent immunolocalization on cross sections in a fixative solution containing 4% formaldehyde (freshly prepared from paraformaldehyde powder, Sigma) and 0.5% glutaraldehyde (Sigma) in a 0.1 M phosphate buffer pH7. Fixation, dehydration and resin infiltration steps were done in a microwave using a PELCO BioWave Pro (Ted Pella, Redding, CA). Fixation was achieved at 150 W, under vacuum (20 Hg) and for 6 min (2'. Followed by a 2'break and a final 2' microwave). Samples were left in the fixative overnight at 4°C and then washed 3 times 3 min (20 Hg, MW 150 W 1' on – 1' break – 1' on). Roots were then aligned and embedded in 1% low-melting Agarose (Calbiochem) and processed through increasing dehydration steps (25%, 50%, 70%, 90%, 96%, 3 × 100% Ethanol, vacuum 20 Hg, MW 150 W 1' on followed by 5' break). Resin infiltration (LR White medium grade, Agar scientific) was achieved through increasing resin concentrations: 33% resin in ethanol 100%, 66% resin in ethanol 100%, and 3 times 100% resin (20 Hg, MW 200 W 2' on – 2' break – 2' on). Polymerization was conducted overnight at 60°C. Semi-thin sections (0.5 μm) were taken with a Leica EM UC7 ultramicrotome. Callose immunolocalization on semi-thin sections was micro-wave assisted and performed as follows: blocking step (BSA 2% in PBS, 1 mL per slide; primary antibody (anti (1→3)-$\beta$-glucan (Biosupplies, AU), 1/200 in BSA 2% in PBS, 500 μL per slide, MW 170 W 2' on – 2' break – 2' on); three washes in BSA 2% in PBS; secondary antibody (Alexa Fluor 488 anti-mouse IGG Thermo-Fisher Scientific, A-11017, 1/400 in BSA 2% in PBS (MW 170 W, 1'), 500 μL per slide; three washes in BSA 2% in PBS. Slides were finally mounted in a 1:1 solution of AF1 antifadent (Citifluor) with PBS, containing calcofluor as a cell wall counterstain, and imaged by confocal laser scanning microscopy (Zeiss LSM700).

## Acknowledgements

KJO acknowledges the financial support of the BBSRC. We thank Ilya Belevich for preparing material for SBFSEM and Kirsten Knox and Andrea Paterlini for advice on tracer experiments. We thank Pawel Roszak for providing T2 seeds of *psAPL::icals3m* lines. We thank the Francheschi Microscopy and Imaging center for technical support. This work was supported by National Science Foundation grant IOS-1146500 (MK).

## Additional information

### Funding

| Funder | Grant reference number | Author |
| --- | --- | --- |
| National Science Foundation | 1146500 | Michael Knoblauch |
| Biotechnology and Biological Sciences Research Council | | Karl J Oparka |
| Carlsbergfondet | | Kaare H Jensen |
| Villum Fonden | 13166 | Kaare H Jensen |

The funders had no role in study design, data collection and interpretation, or the decision to submit the work for publication.

### Author contributions

TJR-E, Acquisition of data; Analysis and interpretation of data; Participated in the overall concept of the study; Conducted the experiments on flow velocity, solute unloading and batch unloading; KHJ, Analysis and interpretation of data; Drafting or revising the article; Participated in the overall concept of the study; Carried out modeling; KSH, Acquisition of data; Analysis and interpretation of data; Carried out modeling; BMW, AHH, DLM, Acquisition of data; Analysis and interpretation of data; Performed ultrastructural investigations; JK, Acquisition of data; Analysis and interpretation of data; Conducted experiments on flow velocity, solute unloading and batch unloading; Performed ultrastructural investigations; AGM, Acquisition of data; Analysis and interpretation of data; Performed serial block face investigations; DP, Acquisition of data; Analysis and interpretation of data; Conducted the grafting experiments; Conducted the callose induction experiments; DY, SO, MB, Acquisition of data; Analysis and interpretation of data; Conducted the callose induction

experiments; RS, J-YL, Acquisition of data; Contributed unpublished essential data or reagents (provided pCalS8:ER-YFP transgenic lines); YH, Conception and design; Analysis and interpretation of data; Drafting or revising the article; Participated in the overall concept of the study; MK, Conception and design; Acquisition of data; Analysis and interpretation of data; Drafting or revising the article; Designed the overall concept of the study; Conducted the experiments on flow velocity, solute unloading and batch unloading; Performed ultrastructural investigations; Wrote the manuscript with participation of the other authors; KJO, Conception and design; Acquisition of data; Analysis and interpretation of data; Drafting or revising the article; Performed serial block face investigations; Wrote the manuscript with participation of the other authors

## Author ORCIDs

Timothy J Ross-Elliott, http://orcid.org/0000-0002-0991-9588

Kaare H Jensen, http://orcid.org/0000-0003-0787-5283

Brittney M Wager, http://orcid.org/0000-0001-9679-659X

Jan Knoblauch, http://orcid.org/0000-0002-8952-3961

Alexander H Howell, http://orcid.org/0000-0001-6735-0660

Alexander G Monteith, http://orcid.org/0000-0003-1731-0446

Dawei Yan, http://orcid.org/0000-0001-8256-0279

Sofia Otero, http://orcid.org/0000-0001-9409-8544

Michael Knoblauch, http://orcid.org/0000-0003-0391-9891

Karl J Oparka, http://orcid.org/0000-0002-8035-5076

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

# Appendix

## Phloem unloading model

The process of phloem unloading is mediated by plasmodesmata (PD) pores that link phloem sieve elements (SE) and neigboring phloem pole pericycle cells (PPP). Transport of solutes out of the phloem occurs by a combination of advection (bulk flow) and diffusion through PD (*Appendix 1—figure 1a*). The contribution from each mechanisms depends on the PD properties as well as the SE ability to maintain pressure and concentration gradients relative to the PPP.

In this supplement, we consider different unloading scenarios and estimate the pressure differentials required to drive phloem unloading by bulk flow in each case: First, we consider unloading by bulk flow through simple PD. Next, the pressure differential required to drive unloading through funnel PD is evaluated. Finally, we discuss the role of diffusive transport through through both simple and funnel PD, and aquaporin-mediated unloading of water. From these results, we deduce the most probable mode of phloem unloading able to account for our experimental observations.

## Mathematical model of phloem unloading

Phloem unloading from SE to PPP is facilitated by two types of PD (*Appendix 1—figure 1b*). Simple PDs are annular pores of inner radius $a_i$, slit width $w$, and length $t$. Funnel PDs taper from the SE towards the PPP from a wide approximately circular opening of radius $e$. The parameters used in the following calculations are given in *Appendix 1—table 1*.

**Appendix 1—table 1.** Parameters used in unloading transport calculations.

| Parameter | Symbol | Value |
|---|---|---|
| Sap flow speed | $u$ | 23 $\mu$m/s |
| Sieve element radius | $r$ | 1.8 $\mu$m |
| Length of unloading zone | $L$ | 350 $\mu$m |
| Desmotubule radius | $a_i$ | 7.5 nm |
| Cytoplasmic sleeve width | $w$ | 2.8 nm |
| PD outer radius | $a_0 = a_i + w$ | 10.3 nm |
| PD funnel radius | $e$ | 75 nm |
| Cell wall thickness/PD length | $t$ | 500 nm |
| Phloem sap osmotic potential | $c$ | 500 mM |
| Phloem sap viscosity | $\eta$ | 1.7 mPas |
| Phloem sap density | $\rho$ | $10^3$ kg/m$^3$ |
| Diffusivity of sucrose | $D$ | $D = 5 \times 10^{-10}$ m$^2$/s |
| Cell membrane permeability (see Table 3.1 of *Kramer and Boyer [1995]*) | $L_p$ | $5 \times 10^{-14}$ m/s/Pa |
| Number of PD | $N$ | |
| – Bulk flow through simple PD | | 240 |
| – Bulk flow through funnel PD at PSE/PPP-interface | | 216 |
| – Diffusive unloading through simple PD | | 240 |
| – Diffusive unloading through funnel PD | | 216 |

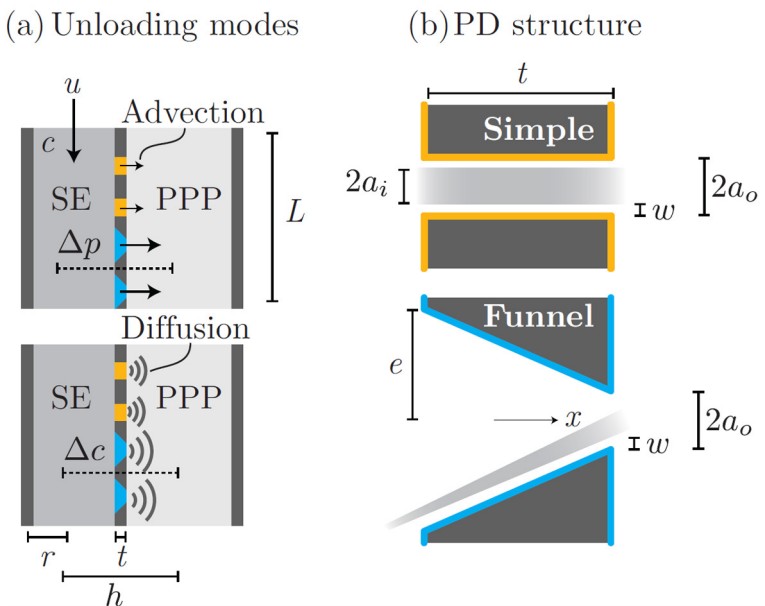

**Appendix 1—figure 1.** Phloem unloading model. Parameters used in calculations for advective and diffusive modes (a): bulk flow and diffusion through PD types (b): simple and funnel. See *Table 1* for parameter descriptions and values.

To ascertain the feasibility of different unloading scenarios we first compute the total amount of sugar unloaded from a single phloem file. Using measured values of the flow speed $u$ and SE radius $r$ yields a flow rate of

$$Q = \pi r^2 u = 2.3 \times 10^{-16} \text{ m}^3/\text{s}. \tag{1}$$

The sugar current, and the amount to be unloaded, is thus

$$I = cQ = 1.2 \times 10^{-13} \text{ mol/s} \tag{2}$$

where $c$ is the sap sugar concentration.

## Unloading by bulk flow through simple PD

Assuming that all liquid must leave the SE via simple PD, the flow rate per PD is $Q_1 = Q/N = 9.8 \times 10^{-19} \ m^3/s$, while the maximum flow speed through each PD active in unloading is $u_{\text{PD}} = Q/(NA_1) = 6.2 \ mm/s$. Here, $N = 240$ is the total number of PD and $A_1 = \pi(a_o^2 - a_i^2) = 1.6 \times 10^{-16} \ m^2$ is the conductive area of one PD with inner radius $a_i = 7.5 \ nm$ and outer radius $a_o = a_i + w = 10.3 \ nm$. The characteristic PD Reynolds number is $Re = \rho u_{\text{PD}} w/\eta = 1.0 \times 10^{-5}$, with viscosity $\eta$ and density $\rho$. In the most general case, flow resistance through the PD has two contributions at low Reynolds numbers: internal viscous resistance to flow in the gap between the plasma membrane and the desmotubule - and - external viscous resistance associated with effects near the pore entrance. This leads to an expression for the pressure drop with two terms

$$\Delta p = \Delta p_{\text{int}} + \Delta p_{\text{ent}}. \tag{3}$$

Both pressure drops depend on details of the PD geometry, the nature of the desmotubule, and on the shape of the entrance region (*Hasimoto, 1958*; *Blake, 1978*;

*Kramer and Boyer, 1995a*; *Jensen et al., 2012*). However, order-of-magnitude estimates can be obtained by assuming laminar Stokes flow inside the PD, and that the entrance region is free from physical obstructions.

Consider the case of unloading by bulk flow through $N$ simple PD. The pressure drops in *Equation (3)* can be estimated by assuming flow throught a straight slit of width $w$ and length $t$. In this case, the two pressures are (*Hasimoto, 1958*; *Bruus, 2008*).

$$\Delta p \simeq \underbrace{12 \frac{\eta u_{pd} t}{w^2}}_{\text{internal resistance}} + \underbrace{\frac{32}{\pi} \frac{\eta u_{pd}}{w}}_{\text{entrance resistance}} = 8.10 \times 10^6 \, \text{Pa} + 4 \times 10^4 \, \text{Pa} = 8.14 \, \text{MPa}. \quad (4)$$

This pressure difference is relatively large, and most likely practically unfeasible, although we emphasize that this is an order-of-magnitude estimate. It is, however, probable that any objects (e.g. large proteins), which are present in the PD, will lead to an increase in $\Delta p$.

## Unloading by bulk flow through funnel PD

We proceed to consider unloading through funnel PD, assuming $N = 216$ are of this type. Since funnel PD posses a relatively wide opening facing the SE, we expect that the internal resistance of these is smaller than that computed for the straight narrow pores (*Equation 4*). There are two possible scenarios depending on the location of the desmotubule. First, we neglect the influence of the desmotubule, and consider flow in an open tapering cylinder. In the lubrication approximation, the hydraulic resistance per unit length of a cylinder of radius $r$ is $R'_c = 8\eta/(\pi r^4)$. The pressure drop across a cylinder tapering linearly in radius from $r(x=0) = e$ to $r(x=t) = a_o$ is thus approximately

$$\Delta p_{\text{int}} \simeq Q_1 \int_0^t R'_c \, \mathrm{d}x = Q_1 \frac{8\eta}{\pi} \int_0^t \frac{\mathrm{d}x}{(e - (e - a_o)x/t)^4} = Q_1 \frac{8\eta}{3\pi} \frac{(e^2 + a_o(e + a_o))t}{e^3 a_o^3} = 1.1 \times 10^4 \, \text{Pa} \quad (5)$$

Second, we take into account the presence of the desmotubule and consider flow in the annular space between two concentric cylinders of inner radius $r_1$ and outer radius $r_2$. The hydraulic resistance per unit length is (*Pozrikidis, 2011*)

$$R'_a = \frac{8\eta}{\pi} \frac{1}{(r_2^2 - r_1^2)\left[r_2^2 + r_1^2 - \frac{(r_2^2 - r_1^2)}{\log r_2/r_1}\right]} \quad (6)$$

The pressure drop for $r_1 = a_i$ and $r_2(x) = e - (e - a_o)x/t$ is

$$\Delta p_{\text{int}} \simeq Q_1 \int_0^t R'_a \, \mathrm{d}x = 1.7 \times 10^5 \, \text{Pa} \quad (7)$$

The external entrance resistance is given by the expression in *Equation (4)*

$$\Delta p_{\text{ent}} = \frac{32}{\pi} \frac{\eta u_{pd}}{w} = 4.3 \times 10^4 \, \text{Pa}. \quad (8)$$

In summary, a total pressure drop of

$$\Delta p = \Delta p_{\text{int}} + \Delta p_{\text{ext}} = 0.05 \, \text{MPa} \quad \text{to} \quad 0.2 \, \text{MPa} \quad (9)$$

is required to drive unloading through funnel PD depending on the detailed location of the desmotubule. A cell-to-cell pressure differential of this magnitude does not appear

unreasonable. Finally, we note that the pressure required for bulk unloading via funnel PD is at least 40 times smaller than that found for simple PD (**Equation 4**).

## Unloading by diffusion through simple PD

We proceed by considering the contribution to unloading from diffusion. A sugar molecule faces three obstacles when diffusing from SE to PPP (**Bret-Harte and Silk, 1994**). First, it must traverse the SE cell cytoplasm, i.e. cover a distance of a few microns. Second, it has to pass through the narrow constriction of the plasmodesmata. Finally, it must diffuse throughout the PPP cytoplasm.

We begin by considering the situation in the absence of PDs, i.e. assuming that the entire cell wall area is perfectly conductive. In that case, the unloading rate is

$$I_0 = AD\frac{\Delta c}{h}, \tag{10}$$

where $\Delta c$ is the SE-to-PPP sugar concentration difference and $h = 10~\mu m$ is the sum of the SE and PPP diameter and $A$ is the surface area of the SE. We assume here that $A = 2\pi rL$, where $L = 350~\mu m$ is the length of the unloading zone. The parameters $D = 5 \times 10^{-10}~\mathrm{m^2/s}$ is the diffusion coefficient for sucrose. Equating the diffusive current in **Equation (10)** to the total incoming sap (**Equation (2)**) gives a required concentration gradient of

$$\Delta c = \frac{Qch}{AD} = 0.59~\mathrm{mol/m^3} = 0.59~\mathrm{mM} \tag{11}$$

To ascertain the importance of the PDs in diffusive unloading, we consider opposite of the previous situation, i.e. the case where the sole resistance to diffusion is provided by the PDs. The diffusion rate through a short isolated pore of radius $a_o$ on a flat surface is governed by Stefans diameter law (**Berg, 1993**)

$$I_S = 2NDa_o\Delta c, \tag{12}$$

By equating **Equation (12)** to **Equation (2)** we find

$$\Delta c = \frac{Qc}{2NDa_o} = 21.6~\mathrm{mol/m^3} = 21.6~\mathrm{mM} \tag{13}$$

We proceed to consider the influence of the pore length. The diffusion rate through $N$ pores of radius $a_o$ and length $t$ is given by

$$I_S = ND\pi a_o^2 \frac{\Delta c}{t}, \tag{14}$$

By equating **Equation (12)** to **Equation (2)** we find

$$\Delta c = \frac{Qct}{ND\pi a_o^2} = 21.6~\mathrm{mol/m^3} = 666.4~\mathrm{mM}. \tag{15}$$

Finally, we investigate the combined effects of cytoplasm and PD resistance to diffusion. We follow Berg (**Berg, 1993**) and write the total diffusive current as

$$I = \frac{\Delta c}{R} = \frac{\Delta c}{R_{\mathrm{cyt}} + R_{\mathrm{PD}}^{(1)} + R_{\mathrm{PD}}^{(2)}} \tag{16}$$

where $R$ is the combined diffusion resistance. The three terms in the denominator are $R_{pd}^{(1)} = 1/(2NDNa_o)$ (the diffusion resistance of the PD opening) $R_{pd}^{(2)} = t/(ND\pi a_o^2)$ (the diffusion resistance of the PD interior) and $R_{\mathrm{cyt}} = h/D$ (the diffusion resistance of the cell cytoplasm), corresponding to **Equations (10) and (12)**. The total unloading rate is thus

$$I = \frac{\Delta c}{\frac{h}{D} + \frac{1}{2DNa_o} + \frac{t}{ND\pi a_o^2}} = AD\frac{\Delta c}{h}\frac{1}{1 + \frac{1}{2\rho a_o h}\left(1 + \frac{2}{\pi}\frac{t}{a_o}\right)} \tag{17}$$

where $\rho = N/A$ is the PD density. This leads to an expression for $\Delta c$ of the form

$$\Delta c = \frac{Qch}{AD}\left(1 + \frac{1}{2\rho a_o h}\left(1 + \frac{2}{\pi}\frac{t}{a_0}\right)\right) = 688.6 \ \mathrm{mol/m^3} = 688.6 \ \mathrm{mM} \tag{18}$$

## I.4 Unloading by diffusion through funnel PD

For funnel PD, the cell and pore opening resistances are similar to those compute above

$$R_{\mathrm{cyt}} = \frac{h}{DA} \tag{19}$$

$$R_{\mathrm{PD}}^{(1)} = \frac{1}{2NDa_o} \tag{20}$$

Along the pore length, we take into account the varying cross-sectional area

$$R_{\mathrm{PD}}^{(2)} = \frac{1}{\pi ND}\frac{1}{ND}\int_0^t \frac{\mathrm{d}x}{[e - (e - a_o)x/t]^2} = \frac{1}{\pi ND}\frac{1}{ND}\frac{t}{a_o e}. \tag{21}$$

This leads to a current of

$$I = AD\frac{\Delta c}{h}\frac{1}{1 + \frac{1}{2\rho a_0 h} + \frac{t}{\pi \rho a_o e h}} = AD\frac{\Delta c}{h}\frac{1}{1 + \frac{1}{2\rho a_0 h}\left(1 + \frac{2}{\pi}\frac{t}{e}\right)} \tag{22}$$

Finally, this leads to an expression for the concentration difference required to drive diffusive unloading

$$\Delta c = \frac{Qch}{AD}\left(1 + \frac{1}{2\rho a_o h}\left(1 + \frac{2}{\pi}\frac{t}{e}\right)\right) = 276 \ \mathrm{mol/m^3} = 276 \ \mathrm{mM}. \tag{23}$$

## Unloading of water in purely diffusive unloading

If unloading occurs exclusively by diffusion through PDs, the water contained in the phloem sap must leave the SE by traversing the cell membrane. To estimate the pressure differential required to facilitate this, we assume that the volume flow rate $Q$ move uniformly across the membrane of area $A$ (surface area of SEs). Taking $L_p$ as the permeability of the membrane leads to

$$Q = AL_p \Delta p_{\mathrm{memb}}. \tag{24}$$

It follows that the pressure is given by

$$\Delta p_{\mathrm{memb}} = \frac{Q}{AL_p} \tag{25}$$

The magnitude of the permeability varies considerably between cell types, with values in the range from $L_p \simeq 10^{-14}$ m/s/Pa to $10^{-12}$ m/s/Pa reported by Kramer and Boyer (see Table 3.1 of *Kramer and Boyer [1995]*). This leads to pressures in the range

$$\Delta p_{\mathrm{memb}} = 0.059 \; \mathrm{MPa} \quad \mathrm{to} \quad 5.9 \; \mathrm{Mpa}. \tag{26}$$

Ross-Elliott *et al*. eLife 2017;6:e24125. DOI: 10.7554/eLife.24125

