## [Decision Letter]

Thank you for submitting your article "Phloem unloading in Arabidopsis roots is convective and regulated by the phloem-pole pericycle" for consideration by *eLife*. Your article has been reviewed by three peer reviewers, and the evaluation has been overseen by Christian Hardtke as the Reviewing and Senior Editor. The following individuals involved in review of your submission have agreed to reveal their identity: Ulrich Z Hammes (Reviewer #1); David E Salt (Reviewer #2).

The reviewers have discussed the reviews with one another and the Senior Editor has drafted this decision to help you prepare a revised submission.

As you can see from the reviews below, all three reviewers appreciate your study and believe that your careful mapping of the root phloem unloading process and the identification of a new, specialized plasmodesmata type in this process are an exciting advance in the field. They also have a number of suggestions for further improvement of the manuscript, which we would like you to address one by one to the best of your capacity in a revised version.

Reviewer #1:

The manuscript "Phloem unloading in Arabidopsis roots is convective and regulated by the phloem-pole pericycle" by Ross-Elliot and colleagues presents a breakthrough in our current understanding of phloem unloading- an agronomically important process. This manuscript addresses (and answers) a major question that arose from a previous manuscript of this (now extended) consortium which was published in *eLife* last year: how is phloem unloading achieved if the pressure gradient between the phloem and the surrounding tissue in sinks is low and the high-pressure manifold model that was believed to be working in roots does not work. Additionally, the authors found that phloem unloading occurs in a specialized cell type which is different from the tissue that was – so far – thought to be the major player in this context. An interesting consequence from this paper is also that it shows that possibly/likely many of the macromolecules, a very "hot" topic that is currently discussed controversially, found in the phloem represent "garbage" rather than signaling molecules. Last not least: I spent a while going through the literature: This is the first study that provides data on the anatomy and (sub)cellular organization of the unloading zone which led to the discovery of a new type of plasmodesmata (PD).

Taken together, I believe this paper contains many exciting findings that can be published pretty much without major changes.

Reviewer #2:

In this manuscript the authors provide a wealth of elegant experimental data to help define the process of phloem unloading in roots of *Arabidopsis thaliana*.

I found the conclusions developed by the authors compelling and well supported by the data.

By combining confocal imaging of both GFP-tagged protophloem sieve elements (PSEs) and the fluorescent phloem mobile dye Esculin (which does not accumulate in vacuoles after unloading which is an advantage over carboxyfluorescein) the authors convincingly define the first transport competent PSE in a given file of PSE cells in the root. The authors use fluorescence recovery after photobleaching (FRAP) to measure phloem velocity and observe a reduction in this velocity in the phloem unloading zone. Using the first PSE identified as PSE-zero as a marker for the end of the unloading zone and flow rate reduction as a precise measure of phloem unloading the authors are able to precisely define the extent of the phloem unloading domain. A significant achievement. Strong evidence is presented using fluorescently tagged proteins and confocal imaging that proteins are unloaded in 'batches' and not as a steady stream. Furthermore, using cell-type specific expression of callose synthase the authors establish that unloading from the PSEs occurs into the phloem pole pericycle cells (PPP) and not into the companion cells (CC) as previously proposed. To further investigate the involvement of plasmodesmata (PD) in unloading from the PSEs into the PPP an in depth investigation using serial block-face scanning EM with 3D reconstruction is undertaken. This produces an incredible reconstruction of a file of PSEs and detailed structural information about PD that reveals a new type of 'funnel' PD that connect the PSEs and PPP. This is an exciting discovery. Modelling of phloem unloading is then used to predict that these new funnel PD provide a much more efficient pathway for phloem unloading, which presents a nice potential rational for the existence of such PD.

Overall I found this manuscript to be a fascinating read that describes a study that is rich with significant new insights, observations and testable hypothesis.

Reviewer #3:

In this manuscript, by combination of non-invasive imaging, 3D electron microscopy and mathematical modeling, the authors show that phloem unloading in Arabidopsis root is driven by bulk flow and diffusion ("convection") through plasmodesmata (PD). Imaging supports the idea that phloem pole pericycle (PPP) is a predominant pathway of phloem unloading, rather than companion cells (CCs), supported by the artificial PD closure by callose deposition. Additionally, they also show that most PD interconnecting the protophloem sieve elements (PSEs) and neighboring PPP are shaped like a funnel, which could enable unloading with lower pressure differential between PSE and PPP. Furthermore, they indicate that, during unloading process, large macromolecules are trapped by PPP and restricted in further movement, whereas small solutes are unloaded through PPP and more freely move throughout the root tip, suggesting that PPP may regulate unloading process depending on molecular size, and act as a repository for unloaded macromolecules. Thus, in this study, authors reveal that PPP and its funnel PD act as central players in phloem unloading, which will adds to the regulatory mechanism of phloem unloading and PD function depending on their diverse structures.

This study is mostly descriptive, though the discovery of the phloem unloading route and funnel PDs are important advances. However the mechanism of unloading regulation (e.g. molecular size-dependent regulation, funnel PD formation) is still largely unknown.

Major issues:

I am a little concerned that the data on protein movement is only from 3 proteins, one of which, GFP, is freely mobile, and another, SEOR, is involved in making aggregates, and the third is a foreign protein (aequeorin). There are plenty of native plant fusions available, how would they behave?

The batch unloading is not clear to me, and only shown for GFP in Video 3.

Results section, I don't understand: Once unloading occurred, the flow velocity decreased significantly due to loss of liquid volume during unloading.

Figure 6: Callose staining (I assume in green) appears weaker in the psAPL line (and not all the way around the cells) compared to the pCalS8 line- could this be a reason why the latter blocks transport and not the former? – Indeed, callose could be accumulated at CCs' cell wall by psAPL:iCalS3m, but is the function of PD between SE and CC really defective? To check PD function, authors could examine whether CC-expressed GFP probe can move to surrounding cells.

The references are often very old and recent refs on protein and mRNA movement are not mentioned- the discussion should be more balanced on this and avoid strong opinions on things not investigated in the manuscript, eg: "PPPs may also protect the downstream cells from receiving unwanted,potentially interfering mRNA molecules"

---

## [Author Response]

*Reviewer #3:*

*Major issues:*

*I am a little concerned that the data on protein movement is only from 3 proteins, one of which, GFP, is freely mobile, and another, SEOR, is involved in making aggregates, and the third is a foreign protein (aequeorin). There are plenty of native plant fusions available, how would they behave?*

In the current study, we used three fusion proteins to cover a wide Mr size range. In the recent study by Paultre et al., 2016, it was shown that a host of native fusion proteins leave the terminal protophloem sieve elements and enter a unique ‘post-phloem domain’, identified in the current manuscript as the phloem-pole pericycle. These authors also used fusion proteins with organelle- targeting sequences and so our observations of protein exit from the protophloem are not just restricted to the three fusion proteins reported here.

*The batch unloading is not clear to me, and only shown for GFP in Video 3.*

As explained in the text (subsection “Batch Unloading of Proteins”), we refer to batch unloading when unloading is not continuous, but occurs spontaneously (in batches) into individual cells. This is the case for all probes of the size of GFP and larger. We provide examples in the form of a figure (Figure 5) and two videos (Video 3 and Video 4) which represent the extremes we tested (GFP, 27 kDa and SEOR-YFP 112 kDa). We have numerous videos with other probes (Aequorin GFP, Ubiquitin-GFP etc.), but we do not see a benefit in overloading the figure or adding more videos that basically show the same phenomenon. However, we have mentioned the additional fusion proteins that we tested in the text.

*Results section, I don't understand: Once unloading occurred, the flow velocity decreased significantly due to loss of liquid volume during unloading.*

We have rephrased the sentence to make it more easily understood.

*Figure 6: Callose staining (I assume in green) appears weaker in the psAPL line (and not all the way around the cells) compared to the pCalS8 line- could this be a reason why the latter blocks transport and not the former?*

Fluorescence immunolabelling is suitable for cellular localization of but not usually for quantification as this depends on various factors such as antigen exposure.

*Indeed, callose could be accumulated at CCs' cell wall by psAPL:iCalS3m, but is the function of PD between SE and CC really defective? To check PD function, authors could examine whether CC-expressed GFP probe can move to surrounding cells.*

We agree that it would be helpful to have an additional control. We used the suggested line for our batch unloading studies (pAtSUC2-GFP), but unfortunately it does not provide the results. GFP is constantly supplied to the sieve tube from the companion cells in the metaphloem and in the secondary phloem. Therefore we cannot discriminate between GFP that potentially leaks from protophloem CCs from GFP that is delivered to the root tip from metaphloem CCs. In order to make this experiment as the reviewers suggest, we should engineer a line which is expressing GFP in a CC-specific manner in the protophloem of the unloading zone only. Currently we don’t have such line and establishing such line will go beyond the two-month revision time frame.

*The references are often very old and recent refs on protein and mRNA movement are not mentioned- the discussion should be more balanced on this and avoid strong opinions on things not investigated in the manuscript, eg: "PPPs may also protect the downstream cells from receiving unwanted,potentially interfering mRNA molecules"*

We believe that our references are adequate and that up-to-date references on protein trafficking are indeed included. The suggestion that the PPP may filter unwanted mRNA molecules is simply a hypothesis that could be tested by others in future work.